# EKLF/KLF1 expression defines a unique macrophage subset during mouse erythropoiesis

Kaustav Mukherjee[1,2], Li Xue[1], Antanas Planutis[1], Merlin Nithya Gnanapragasam[1†], Andrew Chess[1], James J Bieker[1,2,3,4]*

[1]Department of Cell, Developmental, and Regenerative Biology, Mount Sinai School of Medicine, New York, NY, United States; [2]Black Family Stem Cell Institute, New York, NY, United States; [3]Tisch Cancer Institute, New York, NY, United States; [4]Mindich Child Health and Development Institute, Mount Sinai School of Medicine, New York, NY, United States

**Abstract** Erythroblastic islands are a specialized niche that contain a central macrophage surrounded by erythroid cells at various stages of maturation. However, identifying the precise genetic and transcriptional control mechanisms in the island macrophage remains difficult due to macrophage heterogeneity. Using unbiased global sequencing and directed genetic approaches focused on early mammalian development, we find that fetal liver macrophages exhibit a unique expression signature that differentiates them from erythroid and adult macrophage cells. The importance of erythroid Krüppel-like factor (EKLF)/KLF1 in this identity is shown by expression analyses in EKLF-/- and in EKLF-marked macrophage cells. Single-cell sequence analysis simplifies heterogeneity and identifies clusters of genes important for EKLF-dependent macrophage function and novel cell surface biomarkers. Remarkably, this singular set of macrophage island cells appears transiently during embryogenesis. Together, these studies provide a detailed perspective on the importance of EKLF in the establishment of the dynamic gene expression network within erythroblastic islands in the developing embryo and provide the means for their efficient isolation.

*For correspondence:
james.bieker@mssm.edu

Present address: †Department of Biological, Geological, and Environmental Sciences, Cleveland State University, Cleveland, OH, United States

Competing interests: The authors declare that no competing interests exist.

## Introduction

Maturation of red blood cells in vivo occurs within specialized niches called 'erythroblastic islands' that consist of a central macrophage surrounded by erythroid cells at various stages of differentiation (*Chasis and Mohandas, 2008*; *Hom et al., 2015*; *Klei et al., 2017*; *Manwani and Bieker, 2008*; *Yeo et al., 2019*). Macrophages aid in providing cytokines for erythroid growth and differentiation, iron for the demands of hemoglobinization, and ultimately phagocytic and DNase functions that consume the extruded, condensed red cell nuclei during enucleation. The island is held together by specifically paired erythroid/macrophage cell surface protein interactions that, in some cases, are cell-type specific (*Chasis and Mohandas, 2008*; *de Back et al., 2014*; *Hampton-O'Neil et al., 2020*; *Manwani and Bieker, 2008*; *Seu et al., 2017*). The ultimate result is a highly effective and efficient means of reticulocyte formation and release.

Recovery from erythropoietic stress is impaired when macrophages are defective (*Chow et al., 2013*; *Jacobsen et al., 2015*; *Liao et al., 2018*; *May and Forrester, 2020*; *Ramos et al., 2013*; *Sadahira et al., 2000*), supporting a physiological role for island macrophages in erythroid biology. Altered islands are associated with poor prognosis of myelodysplastic patients (*Buesche et al., 2016*). Although studies suggest that even an 80% decrease in mouse resident macrophage levels still enables a normal recovery from stress (*Ulyanova et al., 2016*), this response is effectively aided by

differentiation of monocytes to macrophages after recruitment to the splenic red pulp (*Liao et al., 2018*). As steady-state erythropoiesis appears normal in mice with impaired macrophages, the precise role of macrophage in all aspects of erythropoiesis is not fully resolved (*Korolnek and Hamza, 2015*).

Erythroid Krüppel-like factor (EKLF; KLF1 [*Miller and Bieker, 1993*]) is a zinc finger hematopoietic transcription factor that plays a global role in the activation of genes critical for genetic control within the erythroid lineage (reviewed in *Gnanapragasam and Bieker, 2017*; *Siatecka and Bieker, 2011*; *Tallack and Perkins, 2010*; *Yien and Bieker, 2013*). Genetic ablation studies in the mouse show that EKLF is absolutely required for completion of the erythroid program as EKLF-/- embryos are embryonic lethal at E15 due to a profound ß-thalassemia and the low to virtually nonexistent expression of erythroid genes of all categories. At E13.5, they are anemic and their pale fetal liver (FL) is already distinct in EKLF-/- compared to their EKLF+/+ and EKLF+/- littermates. However, EKLF also plays a crucial role in a subset of macrophage cell function, particularly within the erythroblastic island (*Porcu et al., 2011*; *Xue et al., 2014*). Within the island progeny, it directly activates Icam4 in the erythroid compartment and activates Vcam1 in the macrophage compartment (*Xue et al., 2014*). Together, Icam4 and Vcam1 enable a two-pronged adhesive intercellular interaction to occur with their respective integrin partners on the opposite cell type. In the absence of EKLF, these interactions decrease and the integrity of the island is compromised, contributing to the abundance of nucleated, unprocessed cells seen in circulation (*Gnanapragasam et al., 2016*). In addition, loss of Dnase2 expression in the macrophage yields a cell engorged with undigested nuclei that triggers IFNß induction (*Kawane et al., 2001*; *Manchinu et al., 2018*; *Nagata, 2007*; *Porcu et al., 2011*; *Yoshida et al., 2005*).

Independent evidence for EKLF expression in erythroblastic island macrophage has been attained recently by two sets of studies. One study analyzed EpoR+F4/80+ macrophage, which are present in erythroblastic islands and are negative for Ter119, showing that these cells are highly enriched for EKLF (*Li et al., 2019*). In the second study, a pure population of macrophages (*Lopez-Yrigoyen et al., 2018*) derived from a human induced pluripotent stem cell (iPSC) line carrying an inducible KLF1-ER$^{T2}$ transgene (*Yang et al., 2017*) was used to demonstrate that activation of KLF1 in these macrophages altered them to an island-like phenotype as assessed by an increase in expression of erythroblastic island-associated genes and cell surface markers, an increase in phagocytic activity, and an increase in ability to support the maturation and enucleation of umbilical cord blood-derived cells (*Lopez-Yrigoyen et al., 2019*).

The strongest evidence for a specific macrophage subtype in the erythroblastic island comes from the mouse, where F4/80 antigen and Forssman glycosphingolipid expression, but not Mac1 expression, are enriched in these cells (reviewed in *Manwani and Bieker, 2008*). Island macrophages are also larger than peritoneal macrophages and exhibit a high level of phagocytic activity. Although molecular expression differences between macrophage subsets have been observed (*Ginhoux et al., 2016*; *Hom et al., 2015*; *Lavin et al., 2014*; *Seu et al., 2017*), this has not been addressed in the context of early erythroblastic island development in the FL. Given the compelling observations implicating EKLF in island macrophage biology, we characterized the molecular expression of the F4/80+ island macrophages in the developing mouse FL, determined the EKLF-dependent gene expression program in island macrophages using two independent approaches, and then established its role in specifying a unique cellular identity for this cell type by a single-cell analysis approach.

## Results

### Global gene expression in E13.5 FL macrophages reflects both erythroid and macrophage properties

We dissected E13.5 FLs and fluorescence activated cell (FACS)-sorted F4/80+ cells to obtain a pure population of FL macrophages (*Figure 1—figure supplement 1*). Approximately 9% of the total cells in a wild-type FL are F4/80+ (*Figure 1—figure supplement 1A*). The sorted singlets were monitored after cytospin to determine whether they were free of contaminating erythroid cells (*Figure 1—figure supplement 1B*). We found that >95% of the sorted F4/80+ population are single cells and free of any attached or engulfed erythroid cells or nuclei (*Figure 1—figure supplement 1C*). We then used this pure population of F4/80+ FL macrophages to determine their global gene expression profile using RNA-Seq of biological triplicates.

We compared the global gene expression of E13.5 FL F4/80+ macrophages with two sets of gene expression data. One was from primary long-term cultures of extensively self-renewing erythroblasts (ESREs) isolated from FL that can be differentiated to form mature erythroid cells (*England et al., 2011*; *Gnanapragasam et al., 2016*). The second was from adult spleen F4/80+ macrophage (*Lavin et al., 2014*), which is also an in vivo site of erythroblastic islands (*Chow et al., 2013*; *Jacobsen et al., 2015*; *Ramos et al., 2013*). Hierarchical clustering of the gene expression profile from these cell types shows that the FL macrophages cluster closer to differentiating ESREs than to splenic red pulp macrophages (*Figure 1A*), suggesting that the FL macrophages have an early erythroid-like gene expression profile rather than a mature macrophage-like profile. Yet at the same time we find using principal component analysis (PCA) that these cell types cluster separately, indicating that each has a unique identity (*Figure 1B*). Further, we find that for a list of macrophage and erythroid markers (*Figure 1—source data 1*, *Murray and Wynn, 2011*; *Ng and Wood, 2014*), FL macrophages have intermediate expression of both sets of markers compared to ESREs or spleen macrophages (*Figure 1C*). Together, these data suggest that FL F4/80+ macrophages essentially have dual characteristics of erythroid and macrophage-like cell populations in terms of marker expression but still form their own unique subset.

## Cell-type-specific expression of a subset of genes in FL macrophages provides them with a distinct cellular identity

Since our PCA analysis showed that FL macrophages have unique characteristics compared to ESREs and spleen macrophages, we performed k-means clustering of the RNA-Seq datasets of the three cell types (*Figure 1D*). We find a cluster that contains a set of 1291 genes that are almost exclusively expressed in FL macrophages (*Figure 1D* – indicated by flower bracket). Neither ESREs nor spleen macrophages have a similar set of cell-type-specific gene expression as evident from the lack of clusters showing genes only expressed in these cell types (*Figure 1D*). This again suggests that FL macrophages may have a distinct cellular identity and likely possess unique functions compared to other macrophage types. We selected a set of 304 genes that were only expressed in FL macrophages and not in ESREs or spleen macrophages, and refer to them as 'signature genes' (*Figure 1E*, *Figure 1—source data 2*).

To determine whether signature genes are a random subset of genes or whether they indeed have biological significance with respect to FL macrophage function, we performed gene ontology (GO) analysis and filtered the results down to the unique GO terms using REVIGO (*Supek et al., 2011*; *Supplementary file 1*). We find that the signature genes are involved in four major biological processes: circulatory system development, tube development (vasculature development), locomotion and motility, negative regulation of blood coagulation, and cell adhesion (*Supplementary file 1*). Of these, cell adhesion between erythroblast island macrophages and developing erythroblasts during erythropoiesis is known to be an important function of a subset of FL macrophages (*Xue et al., 2014*). The additional GO categories point to novel biological or developmental roles for FL macrophages.

## Loss of EKLF leads to significantly altered gene expression in F4/80+ FL macrophages

As a prelude to analyzing the effects of EKLF on F4/80+ macrophage, we directly verified EKLF protein expression and find that it is expressed in the F4/80+ macrophage as judged by immunofluorescence (*Figure 2A*). Consistent with our previous data, not all F4/80+ cells are EKLF+, and vice versa (*Figure 2A, B*). Additional support for macrophage specificity of EKLF expression comes from a published RNA-Seq analyses of an extensive series of staged, sorted cells in the FL (*Mass et al., 2016*). Mature macrophage cells (ckit-/CD45+/F480+/AA4.1-/CD11b+) do not exhibit EKLF expression in FL at E10.25; however, EKLF expression 6 hr later in the FL is apparent (E10.5) and robust by E12.5, where it remains high until E18.5, dropping off considerably until it is not detectable at postnatal stages in the liver (*Figure 2C*). As a positive control, Adgre1 (F4/80) is expressed in all samples (*Figure 2D*). As a negative control, EKLF is not expressed in any other tissue macrophage cell in the same study (all samples from skin, brain, kidney, and lung; *Mass et al., 2016*).

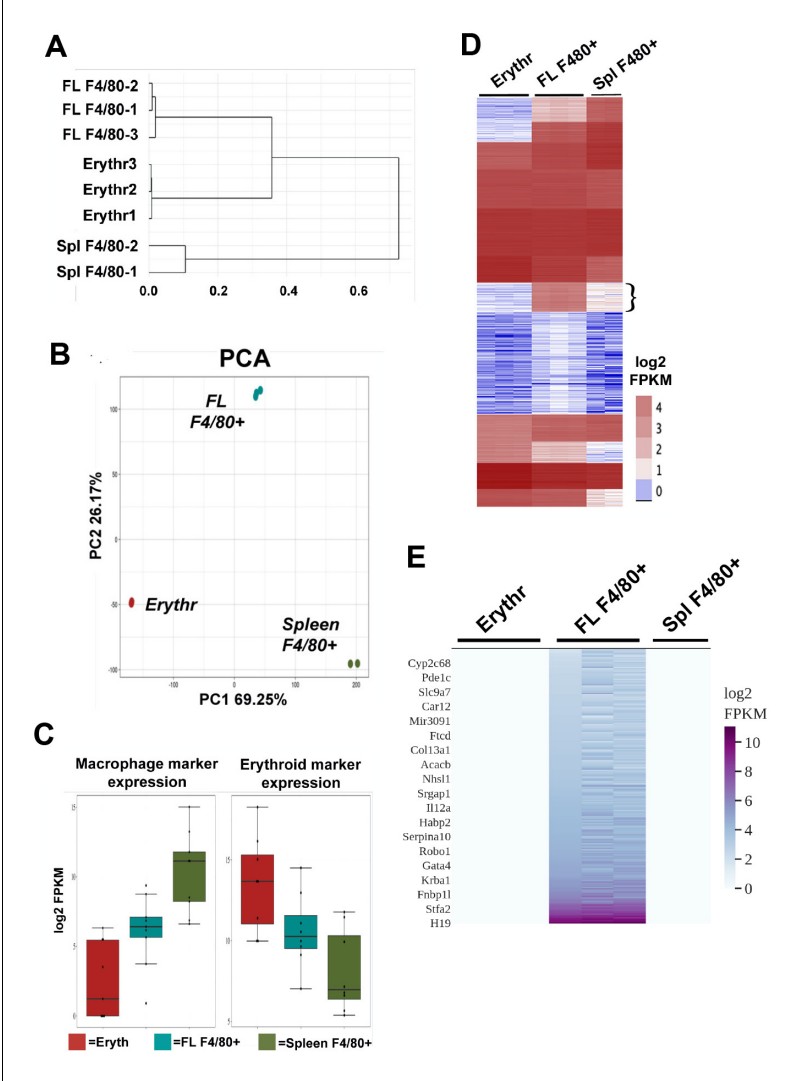

**Figure 1.** Gene expression comparison of fetal liver (FL) F4/80+ macrophages with extensively self-renewing erythroblasts (ESREs; Erythr) and adult spleen (Spl) F4/80+ macrophages showing unique gene expression in F4/80 + FL macrophages. (A) Hierarchical clustering dendrogram using scaled Z-scores based on the expression of the top 10,000 highly expressed genes is shown for individual RNA-Seq biological replicates from each cell type (source data: *Figure 1—source data 1*). (B) Principal component analysis of the cell types is plotted showing principal components 1 and 2 for each biological replicate (source data: *Figure 1—source data 1*). (C) Macrophage-specific or erythroid-specific marker expression in the cell types is shown, with replicates averaged together (source data: *Figure 1—source data 3*). (D) k-means clustering of individual RNA-Seq biological replicates of the different cell types (ESREs, Erythr; fetal liver, FL; spleen, Spl) by log2 FPKM displayed as a heatmap (source data: *Figure 1—source data 4*). Flower bracket indicates the gene cluster with enriched expression in F4/80+ FL macrophages. (E) Heatmap of only the uniquely expressed genes in F4/80+ FL macrophages that define the signature genes of this cell type (source data: *Figure 1—source data 2*). A few representative signature gene names are displayed.

The online version of this article includes the following source data and figure supplement(s) for figure 1:

**Source data 1.** Scaled Z-scores of FPKM values of the top 10,000 highly expressed genes in each cell type shown in *Figure 1*.

**Source data 2.** Expression of signature genes of fetal liver F4/80+ macrophages in each cell type.

**Source data 3.** List of macrophage and erythroid markers and their expression levels in each cell type.

**Source data 4.** Log2 FPKM values of all expressed genes in the cell types shown in *Figure 1*.

**Figure supplement 1.** Isolation of a pure population of F4/80+ E13.5 fetal liver cells by FACS.

*Figure 1 continued on next page*

*Figure 1 continued*

**Figure supplement 1—source data 1.** Quantification of cells isolated by FACS sorting F4/80+ macrophages from E13.5 fetal liver after cytospin.

As a result, we used FACS-sorted F4/80+ FL macrophage from an EKLF-/- mouse and compared its gene expression with wild-type (WT) FL F4/80+ macrophage by RNA-Seq to determine which genes are affected by the loss of EKLF. We observe that there are about half as many F4/80+ FL macrophages in EKLF-/- FL as in WT, suggesting a vital role for EKLF in FL macrophage development (*Figure 3A*; compare to *Figure 1—figure supplement 1A*). Using k-means clustering of the RNA-Seq data, we find the predominant effect is that genes are downregulated in the EKLF-/- FL macrophages (*Figure 3B*). This is consistent with the role of EKLF as a transcriptional activator (*Miller and Bieker, 1993*). We performed differential gene expression analysis using DESeq2 and found that a set of 1210 genes are significantly downregulated in the EKLF-/- FL macrophages (*Figure 3C*, *Supplementary file 2*). Using REVIGO analysis, we find that among others many of the downregulated genes are involved in cell–cell adhesion (*Table 1*).

## EKLF-expressing F4/80+ FL cells are a functionally distinct population from EKLF- F4/80+ cells based on their gene expression program

In our previous study, we had used a mouse strain derived from embryonic stem cells that contain a single copy of the EKLF promoter directly upstream of a green fluorescent protein (GFP) reporter (pEKLF/GFP) to address whether EKLF might be expressed in both the erythroid cell *and* macrophage (*Lohmann and Bieker, 2008*). This promoter/enhancer construct is sufficient to generate tissue-specific and developmentally correct expression in vitro and in vivo (*Chen et al., 1998*; *Lohmann et al., 2015*; *Xue et al., 2004*; *Zhou et al., 2010*); thus GFP onset faithfully mirrors EKLF onset (*Lohmann and Bieker, 2008*). Using this surrogate marker, we had found that ~36% of F4/80+ macrophage singlet cells express EKLF (*Xue et al., 2014*).

Presently, we used FACS to isolate both F4/80+GFP+ (EKLF+) and F4/80+GFP- (EKLF-) subsets and assayed gene expression using RNA-Seq. PCA (*Figure 4A*) and correlation analysis (*Figure 4—figure supplement 1A*) show that the two populations have widely distinct gene expression profiles. Differential expression analysis shows that 2330 genes are enriched in F4/80+EKLF/GFP+ (*Figure 4—figure supplement 1B*, *Supplementary file 3*), with EKLF and Vcam1 among the enriched mRNAs consistent with prior work (*Xue et al., 2014*, *Figure 4B, C*). In addition, we find that Epor mRNA is also enriched in F4/80+EKLF/GFP+ (*Figure 4B, D*). Since Epor+/F4/80+ macrophages form erythroblast islands in bone marrow (*Li et al., 2019*), our data indicates that the same is true for EKLF+ F4/80+ FL macrophages. When we analyze the functional categories of genes significantly enriched in each of the subsets (*Figure 4—figure supplement 1B*), we find that the EKLF/GFP+F4/80+ subset is enriched for genes involved in heme synthesis, iron transport and homeostasis, and myeloid/erythroid differentiation (*Table 2*), functions consistent with those performed by erythroblast island macrophages. In contrast, the genes enriched in EKLF- F4/80+ macrophages are mostly involved in innate and cellular immune responses (*Table 3*), indicating that these are inherently distinct from the EKLF-expressing macrophages in mouse FL.

## EKLF specifies expression of a substantial number of genes including important transcription factors in FL macrophages

Both the above datasets provide us with unique information. The first dataset (*Figure 3*) identifies EKLF-dependent macrophage genes but does not distinguish between EKLF-expressing and EKLF-deficient macrophages in a genetically unaltered state. The second dataset (*Figure 4*) identifies genes with enriched expression in F4/80+ cells where EKLF is also expressed but does not identify EKLF-dependent genes. By comparing the datasets, we can determine which genes have enriched expression in EKLF-expressing macrophages and are also significantly downregulated in EKLF-/-, and therefore truly EKLF-dependent (*Figure 5A*, red box). Overlapping these two independent datasets is an extremely powerful way to parse down the potential direct/indirect genes whose expression is dependent on the presence of EKLF. We find that 504 genes are EKLF-dependent in F4/80

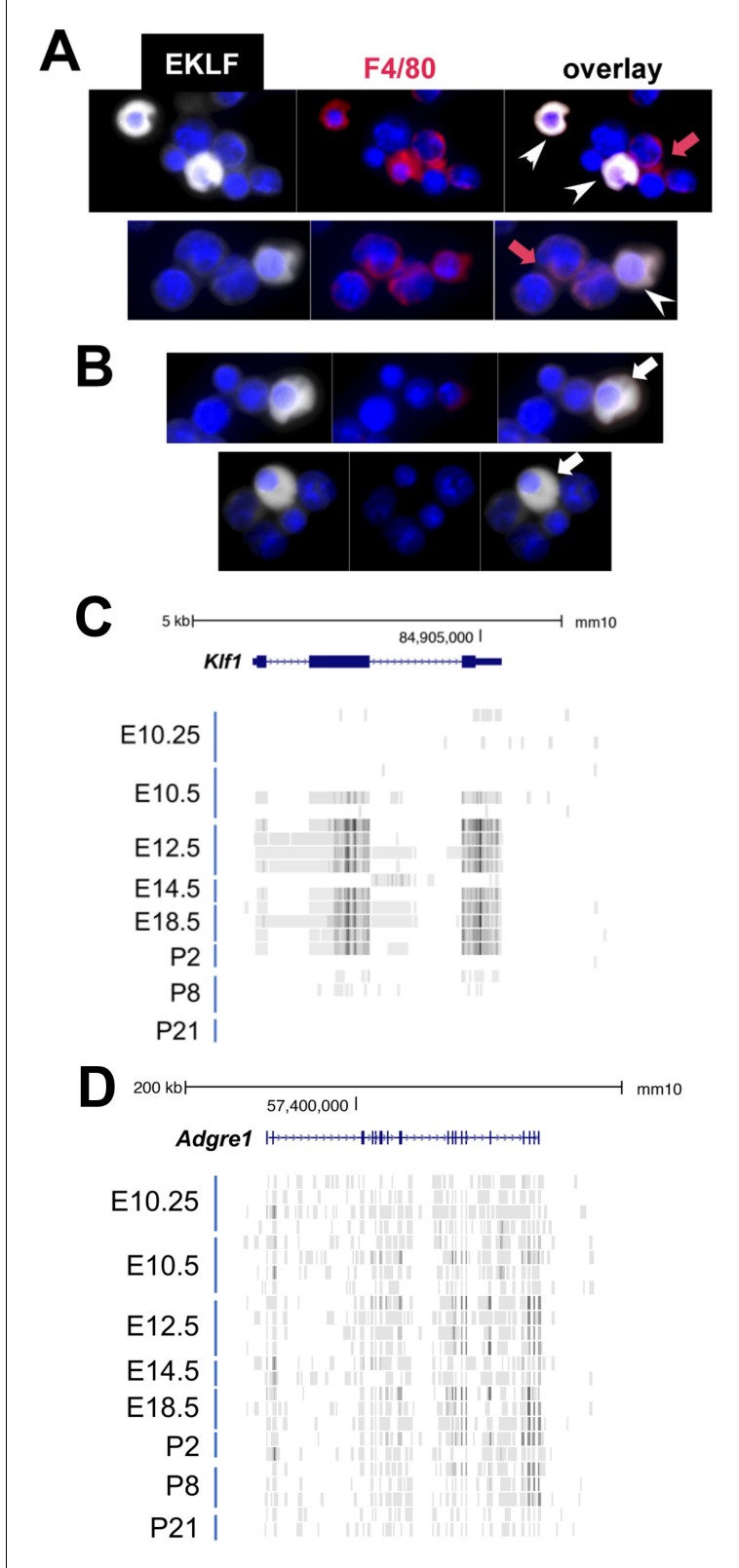

**Figure 2.** EKLF/Klf1 is expressed in fetal liver macrophages during development. (**A**) Immunofluorescence tests with anti-EKLF (white), 4′,6-diamidino-2-phenylindole (DAPI) (blue), and anti-F4/80 (red) antibodies in E13.5 fetal liver cells. (**A**) White arrowheads show coexpression of EKLF and F4/80 proteins in single cells (representative of over 20 EKLF+/F4/80+ cells in this field of 300 cells); red arrow shows that not all F4/80+ cells are EKLF+. (**B**) White

*Figure 2 continued on next page*

*Figure 2 continued*

arrow shows that not all EKLF+ cells are F4/80+ , as expected from the FACS data (cytoplasmic EKLF signal is expected [*Quadrini et al., 2008*; *Schoenfelder et al., 2010*]). (**C**) Collated RNA-Seq data (*Mass et al., 2016*) of sorted macrophage cells from multiple staged embryonic (E) day 10.25–16.5 fetal livers or postnatal (P) day 2–21 livers (ckit-/CD45+/F480+/AA4.1-/CD11b+; n = 24 samples) show transient and abundant Klf1 reads (UCSC Genome Browser). (**D**) Same analysis as (**C**) showing RNA-Seq reads of the gene encoding F4/80 (Adgre1) as a positive control across all samples.

+EKLF+ macrophages, a highly significant number given the size of the datasets (*Figure 5B*, *Figure 5—figure supplement 1A*).

To determine whether these genes may be under EKLF transcription control, we used Centrimo (MEME suite) to analyze the promoters of these 504 genes for TF motifs that are differentially enriched over a background set comprising promoter sequences of the rest of the transcriptome (*Supplementary file 4*). Indeed, we find that Klf1 motifs are overrepresented in these promoters, consistent with the idea that they are EKLF-dependent (*Figure 5C*). In addition, we find that the motifs of transcription factors Klf3, E2f1, E2f4, and Sp4 are significantly enriched (*Figure 5D*) and these TFs are also among the 504 EKLF-dependent genes (*Figure 5E*). This strongly suggests that EKLF, together with Klf3, E2f1, E2f4, and Sp4, may constitute a transcriptional network regulating the distinct gene expression program of FL island macrophages. E2f2 is also EKLF-dependent in F4/

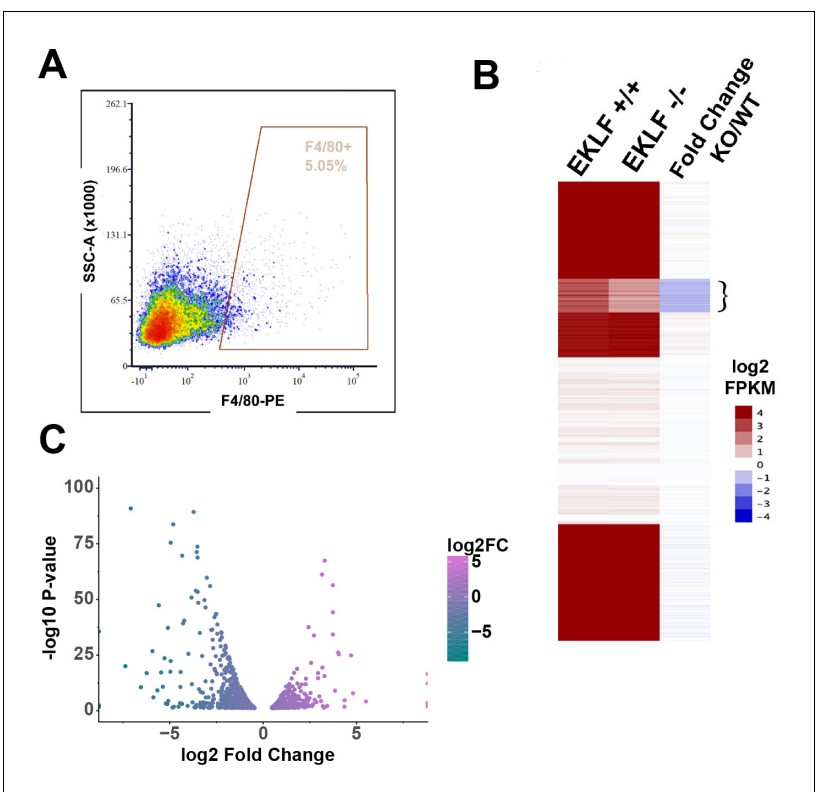

**Figure 3.** EKLF-dependent gene expression in fetal liver (FL) macrophages. (**A**) A representative yield of cells from EKLF-/- FL sorted by F4/80 expression, used for RNA-Seq analysis, is shown (compare to WT yield in *Figure 1—figure supplement 1A*). (**B**) k-means clustering of absolute log2 FPKM of F4/80+ EKLF+/+ and F4/80+ EKLF-/-, and log2 FKPM ratio EKLF-/-(KO)/WT is displayed as a heatmap. Flower bracket indicates downregulated genes. (**C**) Differentially expressed genes in EKLF-/- (KO) compared to WT shown as a volcano plot (source data: *Figure 3—source data 1*).

The online version of this article includes the following source data for figure 3:

**Source data 1.** Differential expression test results obtained from DESeq2 using the RNA-Seq data from EKLF+/+ and EKLF-/- fetal liver F4/80+ macrophages.

**Table 1.** Summary of significant GO terms for the subset of genes significantly downregulated in EKLF-/- vs WT.

| Term_ID | Description | Frequency (%) | log10 p-value |
|---|---|---|---|
| GO:0006464 | Cellular protein modification process | 16.80 | −6.8427 |
| GO:0032502 | Developmental process | 27.72 | −6.6535 |
| GO:0051179 | Localization | 26.83 | −5.5895 |
| GO:0030097 | Hemopoiesis | 3.96 | −5.5005 |
| GO:0048518 | Positive regulation of biological process | 24.84 | −4.1355 |
| GO:0044699 | Single-organism process | 65.98 | −3.6956 |
| GO:0022610 | Biological adhesion | 6.66 | −3.5792 |
| GO:0016043 | Cellular component organization | 27.23 | −3.3699 |
| GO:0008152 | Metabolic process | 51.22 | −3.3045 |
| GO:0071840 | Cellular component organization or biogenesis | 27.98 | −3.2058 |
| GO:0065007 | Biological regulation | 57.48 | −3.1341 |
| GO:0065008 | Regulation of biological quality | 15.62 | −2.4455 |
| GO:0044763 | Single-organism cellular process | 47.39 | −2.4259 |
| GO:0007169 | Transmembrane receptor protein tyrosine kinase signaling pathway | 2.62 | −2.028 |
| GO:0009791 | Post-embryonic development | 0.60 | −1.8994 |
| GO:0098609 | Cell–cell adhesion | 4.30 | −1.7231 |
| GO:0008219 | Cell death | 8.78 | −1.5167 |
| GO:0002376 | Immune system process | 11.16 | −1.4235 |
| GO:0009987 | Cellular process | 75.10 | −1.408 |

80+ macrophages (*Figure 5E*), but its motif is not significantly enriched (*Figure 5—figure supplement 1B*, E-value=0.17), suggesting that E2f2 may not be a critical part of the EKLF transcription network in island macrophages.

The overlap of the datasets (*Figure 5A, B*) suggests that EKLF may regulate the expression of a significant number of other transcription factors in FL macrophages, including Foxo3, Ikzf1, MafK, Nr3c1; cell-cycle E2f factors; and other members of the Klf family (*Figure 5—figure supplement 1C*). This will ultimately be verified by a search of consensus target sequences in putative target genes and by EKLF ChIP. Thus, along with the known transcriptional role of EKLF in erythroid cells, our data is consistent with a global regulatory role for EKLF in the proliferation and development of FL island macrophages.

## Novel EKLF-dependent markers of EKLF+ F4/80+ FL macrophages

Our data has shown that FL macrophages have a distinct cellular identity, with a unique gene expression signature, and that the EKLF+ subset is functionally distinct. We next wished to develop a strategy to isolate the EKLF+ macrophages by finding a novel specific cell surface marker for sorting these cells. We find that of the 304 F4/80+ signature genes (*Figure 1E*), 16 are enriched in F4/80+EKLF/GFP+ macrophages (*Figure 5—figure supplement 2A*) and 32 are downregulated in F4/80+ EKLF-/- macrophages (*Figure 5—figure supplement 2B*). Among these, Adra2b codes for a cell surface adrenergic receptor $\alpha_{2B}$ (*Weinshank et al., 1990*), is highly enriched in F4/80+EKLF/GFP+, and significantly downregulated approximately eightfold in F4/80+EKLF-/- (*Figure 5—figure supplement 2A, B*). We reasoned that Adra2b, along with F4/80+, could be used as an additional marker for EKLF+F4/80+ macrophages. Thus, we determined the proportion of Adra2b and F4/80 expressing cells in E13.5 FLs from EKLF+/+ and EKLF-/- mice using flow cytometry.

Using antibodies against Adra2b and F4/80, we find that whereas only a fraction of Adra2b+ cells are also F4/80+, most F4/80+ FL cells are Adra2b+; however, we also note that the Adra2b+F4/80+ population has a F4/80-hi subpopulation (*Figure 5—figure supplement 2C*, top). This F4/80-hi/Adra2b+ subset is significantly smaller in EKLF-/- (*Figure 5—figure supplement 2C*, bottom),

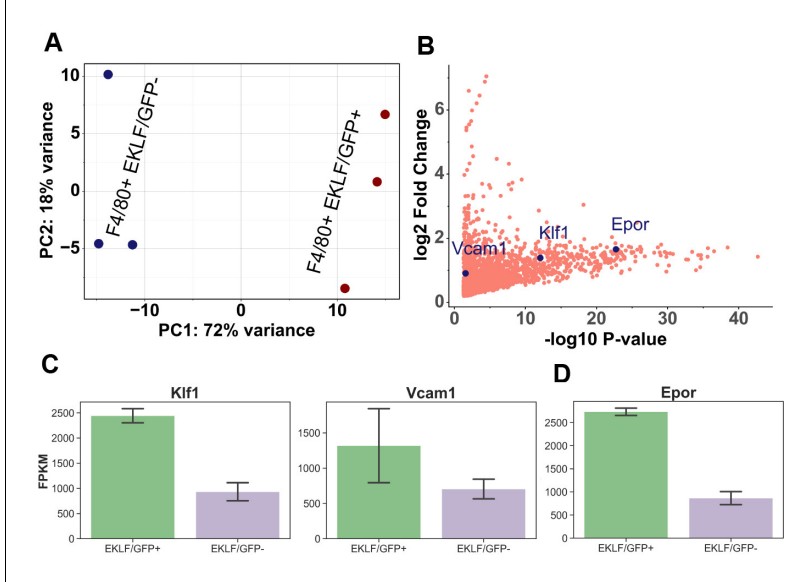

**Figure 4.** Comparison of gene expression in F4/80+ EKLF/GFP+ and F4/80+ EKLF/GFP- fetal liver macrophages. (**A**) Principal component analysis using scaled Z-score based on the expression level of the top 10,000 highly expressed genes from RNA-Seq replicates of F4/80+ EKLF/GFP+ and F4/80+ EKLF/GFP- is plotted with each axis depicting the two major principal components (source data: *Figure 4—source data 1*). (**B**) Scatterplot showing the significantly enriched genes in the F4/80+ EKLF/GFP+ population compared to F4/80+ EKLF/GFP-. Vcam1, Klf1, and Epor are highlighted in blue (source data: *Figure 4—source data 2*). Fragments per kilobase million (FPKM) values of (**C**) EKLF/Klf1 and Vcam1 and (**D**) Epor in the two populations. The online version of this article includes the following source data and figure supplement(s) for figure 4:

**Source data 1.** Coordinates of principal components 1 and 2 corresponding to each replicate of EKLF/GFP+ and EKLF/GFP- RNA-Seq data.
**Source data 2.** Differential gene expression results obtained using DESeq2 from the EKLF/GFP RNA-Seq dataset.
**Figure supplement 1.** Comparison of gene expression in F4/80+ EKLF/GFP+ and F4/80+ EKLF/GFP- fetal liver macrophages.
**Figure supplement 1—source data 1.** Scaled Z-scores of FPKM values of the top 10,000 highly expressed genes in the EKLF/GFP+ F4/80+ RNA-Seq dataset.

consistent with our RNA-Seq observations. This data demonstrates that the F4/80-hi/Adra2b+ population in the FL correlates with EKLF expression in F4/80+ FL cells, suggesting that EKLF+ FL macrophages could be isolated using this strategy.

We used immunofluorescence to directly demonstrate that Adra2b protein is expressed in erythroblastic islands (*Figure 5—figure supplement 2D*). The localization of Adra2b at the surface of the central macrophage cell readily distinguishes it from the more diffuse staining exhibited by F4/80.

## Resolving the cellular heterogeneity in F4/80+ FL macrophages

One critical issue is that FL macrophages are a heterogeneous population of cells, a notion readily apparent from the published literature (*Lee et al., 2018*; *Seu et al., 2017*) and from our own observation that not all F4/80+ cells express EKLF (*Figure 4*). To segregate FL F4/80+ subpopulations and illuminate the role of EKLF in this process, we performed single-cell RNA-Seq on purified F4/80+ FL cells. We used a magnetic bead purification strategy in the presence of Icam4/αv inhibitor peptide (*Xue et al., 2014*) to isolate and maintain healthy F4/80+ cells for single-cell barcoding and library preparation using the Chromium V3 platform (see 'Materials and methods'). Using flow cytometry, we find that about 83% of our purified population is F4/80+ after two rounds of selection (*Figure 6—figure supplement 1*).

Single-cell RNA-Seq confirmed the cellular heterogeneity in the F4/80+ population, with 13 separate clusters of cells after unsupervised dimensionality reduction using the Seurat package (*Butler et al., 2018*; *Stuart et al., 2019*; *Figure 6A*). F4/80+ mRNA (encoded by the Adgre1 gene) is present in all the clusters, although some clusters have higher levels (*Figure 6B*). Additional macrophage markers such as Marco and Vcam1 mRNAs are also present in all clusters, whereas the macrophage transcription factor PU.1 (encoded by Spic) is enriched in

**Table 2.** Summary of GO terms for genes significantly enriched in EKLF/GFP+ F4/80+ fetal liver macrophages.

| Term_ID | Description | Frequency (%) | log10 p-valueue | Uniqueness |
|---|---|---|---|---|
| GO:0006778 | Porphyrin-containing compound metabolic process | 0.18 | −12.8484 | 0.777 |
| GO:0051186 | Cofactor metabolic process | 1.60 | −12.2111 | 0.915 |
| GO:0033013 | Tetrapyrrole metabolic process | 0.20 | −11.1538 | 0.869 |
| GO:0051179 | Localization | 26.83 | −8.8728 | 0.994 |
| GO:0034101 | Erythrocyte homeostasis | 0.58 | −8.3027 | 0.786 |
| GO:0006810 | Transport | 20.74 | −8.0986 | 0.952 |
| GO:0051234 | Establishment of localization | 21.48 | −7.6831 | 0.957 |
| GO:0065008 | Regulation of biological quality | 15.62 | −6.8115 | 0.959 |
| GO:0055085 | Transmembrane transport | 5.98 | −6.4975 | 0.945 |
| GO:0042592 | Homeostatic process | 7.64 | −6.1826 | 0.886 |
| GO:0061515 | Myeloid cell development | 0.32 | −5.6997 | 0.83 |
| GO:0048731 | System development | 21.00 | −5.4423 | 0.93 |
| GO:1901564 | Organonitrogen compound metabolic process | 9.12 | −5.1438 | 0.923 |
| GO:0042744 | Hydrogen peroxide catabolic process | 0.07 | −5.0001 | 0.868 |
| GO:0006811 | Ion transport | 7.05 | −4.9442 | 0.946 |
| GO:0048513 | Animal organ development | 15.85 | −4.7331 | 0.926 |
| GO:0008152 | Metabolic process | 51.22 | −4.3756 | 0.997 |
| GO:0044237 | Cellular metabolic process | 45.64 | −4.3162 | 0.937 |
| GO:0055076 | Transition metal ion homeostasis | 0.58 | −4.2923 | 0.831 |
| GO:0007275 | Multicellular organism development | 23.55 | −4.161 | 0.933 |
| GO:0032502 | Developmental process | 27.72 | −4.1078 | 0.994 |
| GO:0048872 | Homeostasis of number of cells | 1.37 | −4.0386 | 0.845 |
| GO:0008643 | Carbohydrate transport | 0.71 | −3.9826 | 0.911 |
| GO:0048878 | Chemical homeostasis | 4.85 | −3.8372 | 0.85 |
| GO:0006796 | Phosphate-containing compound metabolic process | 13.70 | −3.7212 | 0.925 |
| GO:0006793 | Phosphorus metabolic process | 14.00 | −3.5364 | 0.925 |
| GO:0055072 | Iron ion homeostasis | 0.38 | −3.5032 | 0.829 |
| GO:0030099 | Myeloid cell differentiation | 1.70 | −3.3938 | 0.848 |
| GO:0042440 | Pigment metabolic process | 0.30 | −3.2046 | 0.893 |
| GO:0050801 | Ion homeostasis | 3.30 | −3.1372 | 0.843 |
| GO:0048856 | Anatomical structure development | 25.70 | −2.9598 | 0.947 |
| GO:0006820 | Anion transport | 2.43 | −2.8403 | 0.941 |
| GO:0017001 | Antibiotic catabolic process | 0.52 | −2.6912 | 0.876 |
| GO:0098771 | Inorganic ion homeostasis | 3.02 | −2.6573 | 0.839 |
| GO:0019755 | One-carbon compound transport | 0.06 | −2.4416 | 0.9 |
| GO:0019725 | Cellular homeostasis | 3.80 | −2.1148 | 0.81 |
| GO:0071704 | Organic substance metabolic process | 49.01 | −2.0066 | 0.945 |

clusters 0, 1, 2, and 8 (*Figure 6C*). Differential enrichment analysis reveals the mRNAs that are enriched in each cluster (*Figure 6D*, *Figure 6—source data 1*), and we find certain genes with almost exclusive expression in a particular cluster that serve as markers for that cluster (*Figure 6—figure supplement 2*).

It is apparent from these analyses that clusters 0 and 1 have a high overlap in cluster markers (*Figure 6D*), and due to the high expression of macrophage-specific genes (*Figure 6B, C*), these clusters likely are comprised of macrophages. This is also confirmed by GO analysis of the top 100

**Table 3.** Summary of GO terms for genes significantly enriched in EKLF/GFP- F4/80+ fetal liver macrophages.

| Term_ID | Description | Frequency (%) | log10 p-valueue | Uniqueness |
|---|---|---|---|---|
| GO:0002376 | Immune system process | 11.16 | −90.8785 | 0.492 |
| GO:0001775 | Cell activation | 4.73 | −58.5949 | 0.502 |
| GO:0045321 | Leukocyte activation | 4.17 | −57.3276 | 0.435 |
| GO:0001816 | Cytokine production | 2.93 | −56.1898 | 0.522 |
| GO:0040011 | Locomotion | 7.21 | −51.1415 | 0.515 |
| GO:0001817 | Regulation of cytokine production | 2.62 | −48.8035 | 0.477 |
| GO:0006928 | Movement of cell or subcellular component | 7.93 | −47.6096 | 0.474 |
| GO:0006954 | Inflammatory response | 2.89 | −45.4918 | 0.524 |
| GO:0022610 | Biological adhesion | 6.66 | −45.4019 | 0.519 |
| GO:0030334 | Regulation of cell migration | 3.21 | −37.8661 | 0.468 |
| GO:0051707 | Response to other organism | 4.45 | −34.3509 | 0.498 |
| GO:0009607 | Response to biotic stimulus | 4.67 | −34.2713 | 0.514 |
| GO:0030155 | Regulation of cell adhesion | 2.92 | −34.1483 | 0.502 |
| GO:0022603 | Regulation of anatomical structure morphogenesis | 4.20 | −33.7719 | 0.47 |
| GO:0008283 | Cell proliferation | 8.83 | −29.2539 | 0.482 |
| GO:0030036 | Actin cytoskeleton organization | 2.80 | −28.0214 | 0.516 |
| GO:0030029 | Actin filament-based process | 3.14 | −27.3893 | 0.523 |
| GO:0035295 | Tube development | 3.20 | −25.1916 | 0.507 |
| GO:0008219 | Cell death | 8.78 | −24.5897 | 0.468 |
| GO:0070661 | Leukocyte proliferation | 1.41 | −23.5623 | 0.563 |
| GO:0072358 | Cardiovascular system development | 3.13 | −23.0994 | 0.498 |
| GO:0006793 | Phosphorus metabolic process | 14.00 | −22.2385 | 0.449 |
| GO:0044093 | Positive regulation of molecular function | 7.70 | −22.1855 | 0.47 |
| GO:0006897 | Endocytosis | 3.19 | −19.725 | 0.532 |
| GO:0098657 | Import into cell | 0.29 | −19.1913 | 0.625 |
| GO:0050764 | Regulation of phagocytosis | 0.36 | −18.9505 | 0.578 |
| GO:1902533 | Positive regulation of intracellular signal transduction | 4.07 | −18.4109 | 0.439 |
| GO:0051704 | Multiorganism process | 6.53 | −17.9065 | 0.52 |
| GO:0034097 | Response to cytokine | 3.34 | −17.0337 | 0.525 |
| GO:0032940 | secretion by cell | 4.11 | −16.2249 | 0.486 |
| GO:0002699 | Positive regulation of immune effector process | 0.86 | −14.2444 | 0.51 |
| GO:0007167 | Enzyme-linked receptor protein signaling pathway | 4.02 | −14.1906 | 0.48 |
| GO:0001774 | Microglial cell activation | 0.07 | −12.1713 | 0.623 |
| GO:0010942 | Positive regulation of cell death | 2.72 | −11.1617 | 0.486 |
| GO:0097435 | Supramolecular fiber organization | 2.73 | −11.1108 | 0.525 |
| GO:0042592 | Homeostatic process | 7.64 | −10.9553 | 0.477 |
| GO:0035456 | Response to interferon-beta | 0.22 | −10.2302 | 0.627 |
| GO:0008360 | Regulation of cell shape | 0.65 | −10.1463 | 0.547 |
| GO:0042107 | Cytokine metabolic process | 0.55 | −9.76 | 0.605 |
| GO:0050777 | Negative regulation of immune response | 0.61 | −9.7184 | 0.523 |
| GO:0002444 | Myeloid leukocyte-mediated immunity | 0.38 | −9.7099 | 0.588 |

*Table 3 continued on next page*

*Table 3 continued*

| Term_ID | Description | Frequency (%) | log10 p-valuee | Uniqueness |
|---|---|---|---|---|
| GO:0090130 | Tissue migration | 1.14 | −9.6676 | 0.563 |
| GO:0051129 | Negative regulation of cellular component organization | 2.94 | −9.0279 | 0.504 |

markers for these clusters (*Supplementary file 5*). Further, GO analysis of markers for clusters 2 and 3 yields terms compatible with activated macrophage functions (*Supplementary file 5*), and indeed these clusters express genes correlated with activated macrophages such as Csf1r, Dnase2a, and Il4ra (*Figure 6—figure supplement 3A*). In contrast, GO analysis of the top enriched genes for clusters 4, 5, 7, and 8 relate to erythro-myeloid characteristics and heme metabolism (*Supplementary file 6*), with highly enriched markers for these clusters being glycophorin A, α-synuclein, and α-spectrin (*Figure 6—figure supplement 3B*). A search for the terminal erythroid marker Ter119 (Ly76) yields no results in our single-cell sequencing dataset, indicating that perhaps its mRNA is undetectable and that our F4/80+ purification is largely devoid of terminally differentiating erythroid cells. To further support the heterogeneity of expression in these populations, in contrast we find that the mRNA for the constitutively active gene, Gapdh, is uniformly highly expressed in all clusters (*Figure 6—figure supplement 3C*), whereas CD71 (Tfrc) mRNA was expressed at moderate levels in most clusters (*Figure 6—figure supplement 3D*).

## Cellular heterogeneity in EKLF+ F4/80+ FL macrophages and an improved strategy to isolate this population

Our earlier observations from the pEKLF/GFP mice indicated that about 36% of the F4/80+ FL cells express EKLF (*Xue et al., 2014*). EKLF+ expression is detected exclusively in clusters 4, 5, and 7 (*Figure 7A*), and these clusters comprise about 23% of the cells in our dataset. We also find that most of the EKLF+ cells express Epor (*Figure 7B*), consistent with our earlier observations as well as others (*Li et al., 2019*). To further test our previous observations that Adra2b expression correlates with EKLF expression and is found in erythroblast island macrophages (*Figure 5—figure supplement 2D*), we looked for Adra2b expression in single cells. We find specific Adra2b enrichment in cluster 4, thus correlating with some EKLF+ as well as Epor+ cells, albeit the remaining EKLF-expressing clusters 5 and 7 have little Adra2b expression (*Figure 7C*). This indicates high amounts of heterogeneity even within EKLF+F4/80+ macrophages and suggests that Adra2b alone as a marker is not sufficient to enable efficient isolation of EKLF+F4/80+ cells.

This led us to devise an improved strategy to isolate EKLF+F4/80+ FL cells based on cell surface marker expression by searching for mRNAs enriched in EKLF+ clusters 4, 5, and 7 taken together. We find that Add2 (adducin2), Hemgn (hemogen), Nxpe2 (neurexophilin and PC-esterase domain family, member 2), and Sptb (spectrinβ) are specifically enriched in the EKLF+ clusters (*Figure 8A*). Of these, Add2, Nxpe2, and Sptb encode membrane-associated proteins, which would be preferred for antibody-based isolation strategies such as FACS or magnetic bead separation, and thus are attractive candidates for marker-based separation of EKLF+ F4/80+ cells. Although Add2 and Sptb are known to be highly expressed in erythroid cells (*Chen et al., 2009*; *Franco and Low, 2010*; *Gardner and Bennett, 1987*), RNA-Seq data of ckit-/CD45+/F480+/AA4.1-/CD11b+ macrophages derived from staged mice embryos (*Mass et al., 2016*) shows that Add2 and Sptb mRNAs are indeed expressed in mice FL macrophages from E12.5–E18.5 (*Figure 8B*), with a similar developmental onset to that of EKLF (*Figure 2C*). Additionally, when we search for their expression in our F4/80+ EKLF/GFP+ bulk RNA-Seq dataset, all four markers are significantly enriched in F4/80+ EKLF/GFP+ (*Figure 8C*), thus confirming that their mRNA expression correlates with EKLF mRNA expression in F4/80+ macrophages. Finally, optimal levels of Add2 and Hemgn expression are also EKLF-dependent since we find that they are significantly downregulated in F4/80+ EKLF-/- cells (*Figure 8D*).

Upon staining E13.5 FL cells with both F4/80 and adducin2, or F4/80 and spectrinβ antibodies, we find that the majority (~88%) of the Add2+ or Sptb+ cells are F4/80- (and presumably erythroid). However, about 25% of all F4/80+ cells are Add2+ or Sptb+ in each case (*Figure 9A*), aligning with our single-cell RNA-Seq observations (*Figure 8A*). We repeated the F4/80+ purification and stained

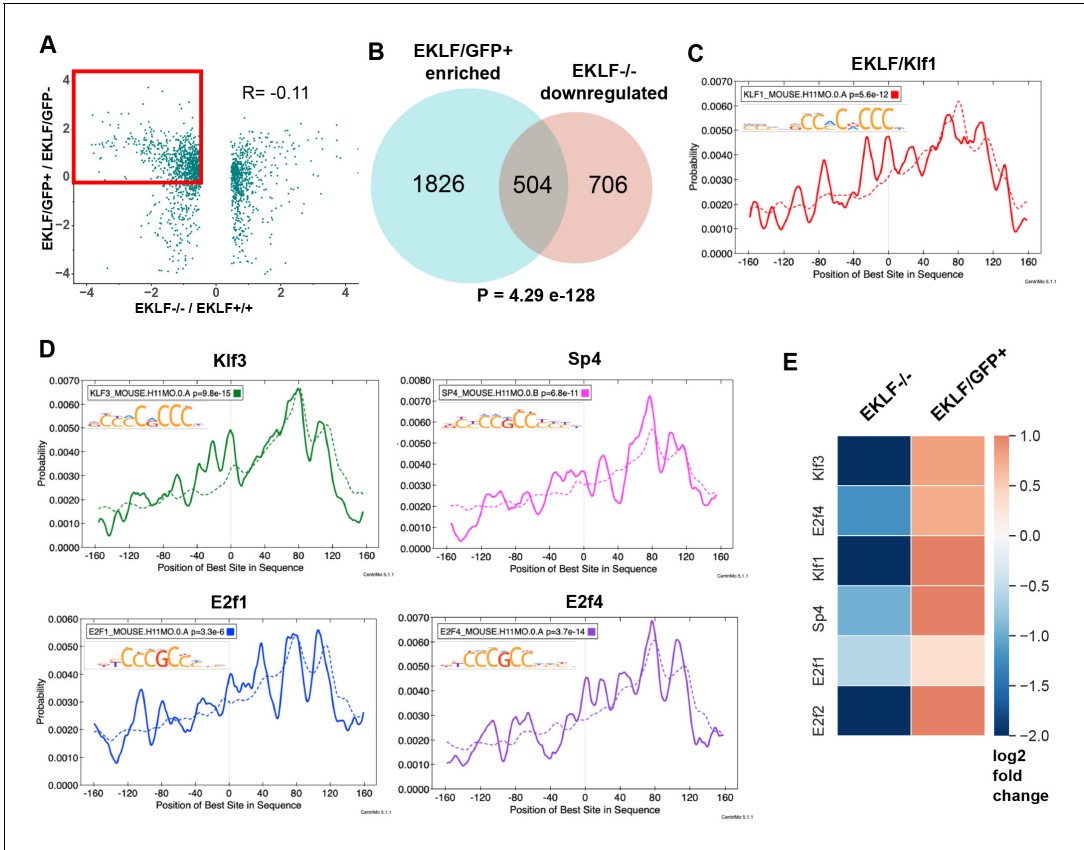

**Figure 5.** EKLF specifies lineage and cell-cycle transcription factors in F4/80+ fetal liver (FL) island macrophages. (**A**) Scatterplot of log2-fold changes in EKLF/GFP+ plotted against EKLF-/-. Red box shows the genes that are common and of interest from both datasets, that is, enriched in EKLF/GFP+ and downregulated in EKLF-/- F4/80+ FL macrophages (source data: *Figure 5—source data 1*). (**B**) Venn diagram showing the number of genes in each category from (**A**). Centrimo analysis of promoters of EKLF-dependent genes showing differential motif enrichment of (**C**) EKLF/Klf1 and (**D**) Klf3, Sp4, E2f1, and E2f4 motifs (source data: *Figure 5—source data 2*, *3*). Dotted line depicts the expected probability of occurrence of the respective motif in the background dataset (see 'Materials and methods'). (**E**) Heatmap showing log2-fold change of expression in EKLF-/- and EKLF/GFP+ of the above EKLF-dependent transcription factors in F4/80+ FL macrophages.

The online version of this article includes the following source data and figure supplement(s) for figure 5:

**Source data 1.** Expression values of differentially expressed genes in EKLF-/- cells vs WT cells compared with their expression in the EKLF/GFP+ dataset.
**Source data 2.** FASTA sequences of the promoters of EKLF-dependent genes.
**Source data 3.** FASTA sequences of the promoters of all genes not included in the EKLF-dependent gene set.
**Figure supplement 1.** EKLF-dependent genes expressed in F4/80+ fetal liver (FL) macrophages.
**Figure supplement 1—source data 1.** Expression values of EKLF-dependent genes.
**Figure supplement 2.** EKLF-dependent signature genes in F4/80+ fetal liver (FL) macrophages.
**Figure supplement 2—source data 1.** Expression values of signature genes from *Figure 1E* that are significantly enriched in EKLF/GFP+ cells.
**Figure supplement 2—source data 2.** Expression of signature genes from *Figure 1E* that are significantly downregulated in EKLF-/- cells compared to WT.

the purified F4/80+ cells for Add2 or Sptb to find that in each case about 24% of the F4/80+ cells are Add2+ or Sptb+ (*Figure 9B*), a proportion resembling the 23% of cells in clusters 4, 5, and 7 where these mRNAs are expressed. To test the possibility that any Add2 and Sptb expression seen in F4/80+ cells was due to residual erythroid contamination in our F4/80+ population, we performed Imagestream analysis. Using the pEKLF/GFP mouse, we stained for F4/80 and Add2, and we find single cells expressing F4/80 and Add2 that are also EKLF/GFP+ (*Figure 9C*). This not only confirms that the Add2 signal is coming from single cells, it also demonstrates visually that Add2 expression in a subset of F4/80+ macrophages correlates with EKLF expression in those macrophages.

Finally, since earlier studies from our group (*Xue et al., 2014*; *Li et al., 2019*) showed that EKLF expression is enriched in macrophages forming erythroblast islands, we isolated erythroblast islands

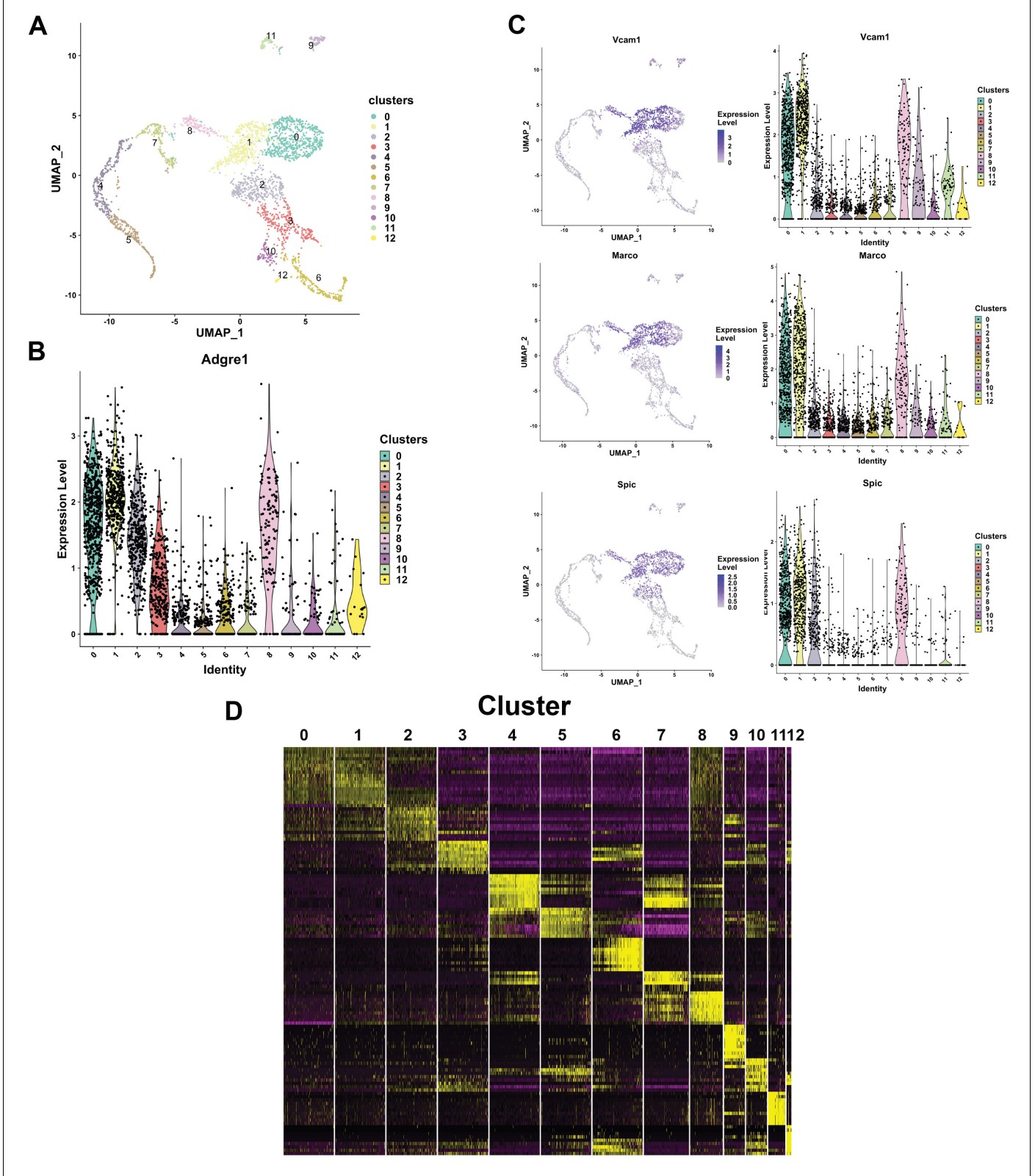

**Figure 6.** Resolving the cellular heterogeneity of E13.5 fetal liver (FL) macrophages using single-cell RNA-Seq. (**A**) Unsupervised clustering using principal component analysis and subsequent U-MAP projections computed and plotted using the R Seurat package for single-cell RNA-Seq of purified E13.5 FL F4/80+ cells. Cluster numbers are indicated on the clusters. (**B**) Violin plot showing the distribution of F4/80 (Adgre1) mRNA expression in the clusters identified in (**A**). (**C**) Feature plots (left panel) showing individual cellular expression superimposed on the cluster, and Violin plots (right)

*Figure 6 continued on next page*

*Figure 6 continued*

showing the distribution of expression in each cluster of macrophage markers Vcam1 and Marco, and the macrophage-specific transcription factor PU.1 (Spic). (D) Differential mRNA enrichment in each cluster plotted as a heatmap, showing putative unique markers of each cluster (source data: *Figure 6—source data 1*). Relative expression levels are indicated by color: yellow=high, black=mid, and purple=low.

The online version of this article includes the following source data and figure supplement(s) for figure 6:

**Source data 1.** Differentially expressed genes associated with each cluster of the single-cell RNA-Seq dataset.
**Figure supplement 1.** F4/80 purity check.
**Figure supplement 2.** Markers for each gene expression-based cluster of cells identified from single-cell sequencing of F4/80+ fetal liver macrophages.
**Figure supplement 3.** Markers of F4/80+ cell clusters with various cell identities.

and tested for Add2 and Sptb protein expression by immunofluorescence. We find high Sptb and Add2 staining in the central macrophage as well as few surrounding erythroid cells (*Figure 9D, E*), indicating that these markers are expressed in erythroblast island macrophages. Thus, Add2 or Sptb can be used as reliable markers to isolate F4/80+ EKLF+ FL island macrophage population for further characterization of their unique properties.

## Discussion

### Identification of a novel cell type in FL macrophage

Although there is overlap among the cell populations, we have shown that E13.5 murine FL F4/80+ macrophages exhibit a distinct expression pattern when compared to adult spleen F4/80+ macrophage, one that is also divergent from that of FL erythroid cells, thus providing them with a discrete cellular identity. Our data suggests the existence of a unique macrophage cell type with novel markers that defines erythroblastic island-associated macrophage. This is perhaps not surprising as there is extensive macrophage heterogeneity (*Lee et al., 2018*; *Paulson, 2019*; *Seu et al., 2017*), and it has been long noted that island macrophage may have a distinctive surface marker expression (*Manwani and Bieker, 2008*).

The unique expression signature exhibited by these cells includes over 300 genes that are functionally involved in positive regulation of developmental processes, particularly cell movement, localization, and adhesion. Our data suggests that establishing a macrophage cell dedicated to maintaining such a unique expression profile makes developmental sense given its role in efficiently aiding the huge demand for red blood cells during early development, specifically within the expanding FL site (*Chasis and Mohandas, 2008*; *Hom et al., 2015*; *Klei et al., 2017*; *Manwani and Bieker, 2008*; *Yeo et al., 2019*).

### Transient nature of a singular, EKLF-dependent FL macrophage population that coincides with the onset of definitive erythropoiesis during mouse embryonic development

The idea of a dedicated island macrophage cell is further supported by the overlap in the single-cell seq and the developmental RNA-Seq expression datasets. These show there is a specific onset of many of the markers of interest that coincide with the peak of EKLF expression in macrophage at E12.5, at the same time as definitive erythropoiesis is occurring in the mouse FL. Strikingly, expression of many of these also dissipates coordinately at later embryonic stages. This may follow from either transient EKLF expression in the macrophage or the transient presence of a population of EKLF-expressing macrophage. Such dynamic regulation has been observed with IL7Rα (*Leung et al., 2019*), but the remarkable coherence of the erythroblastic island macrophage subset in clusters 4, 5, and 7 suggests the existence of a cross-regulatory mechanism that leads to the establishment of a network of genes critical for proper island niche function. Consistent with this, KLF binding motifs are enriched in active macrophage genes (*Gosselin et al., 2014*; *Lavin et al., 2014*) and correlate with binding by other macrophage factors such as CJUN and P65 (*Link et al., 2018*). Our comparative analysis of EKLF-/- and EKLF/GFP+ strongly supports the idea, postulated previously from other studies (*Li et al., 2019*; *Porcu et al., 2011*; *Xue et al., 2014*), that EKLF is a central player in

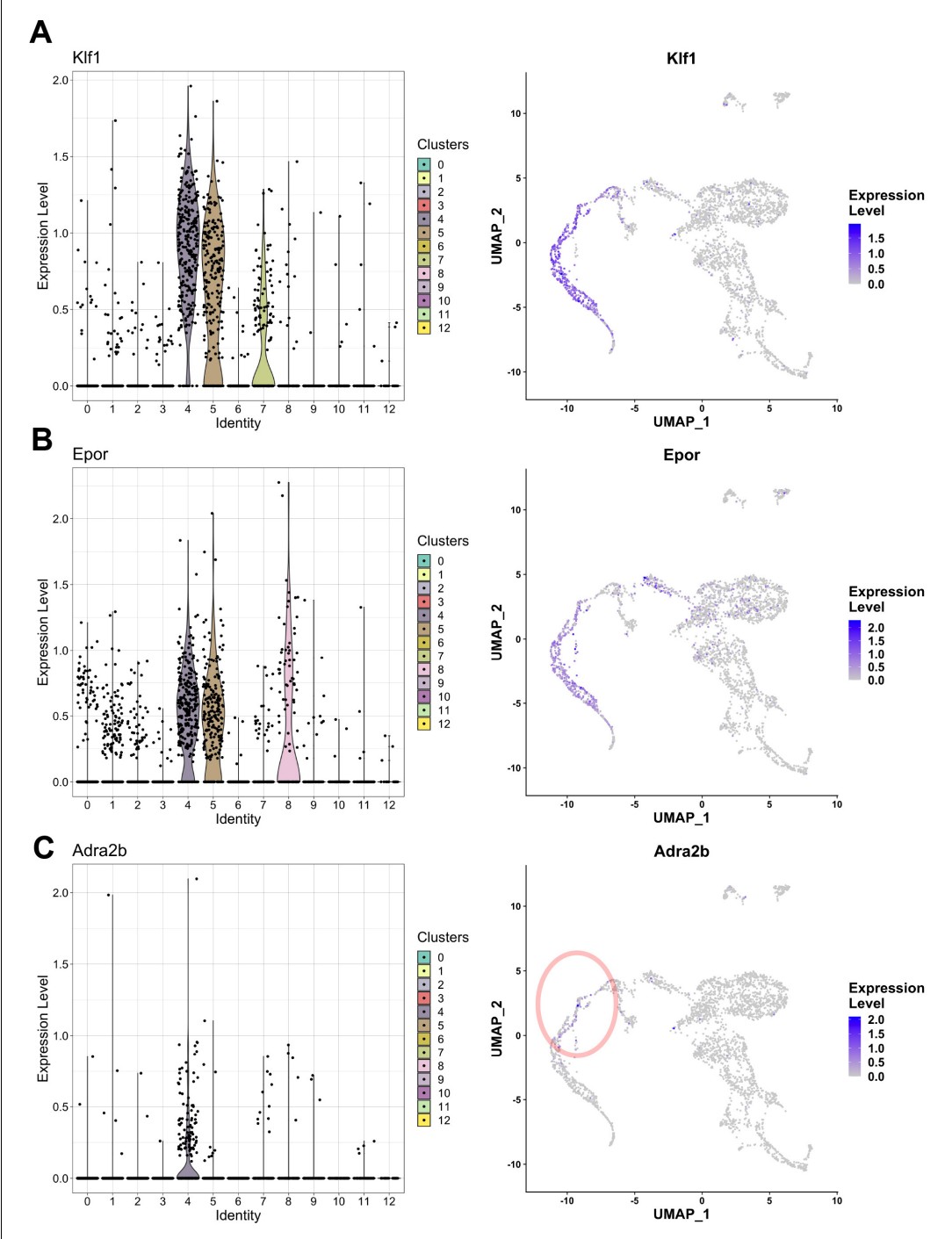

**Figure 7.** EKLF/Klf1-expressing clusters in F4/80+ fetal liver macrophages. Violin plots showing distribution (left) and feature plots (right) showing individual cellular mRNA expression of (**A**) Klf1, (**B**) Epor, and (**C**) Adra2b superimposed on the clusters.

establishing this network at the right time and place in development. Given our studies, the cause of the embryonic lethality in the absence of EKLF could be a combination of impaired erythropoiesis due to the loss of EKLF in developing erythroid progenitors as well as impaired island macrophage function supporting definitive erythropoiesis.

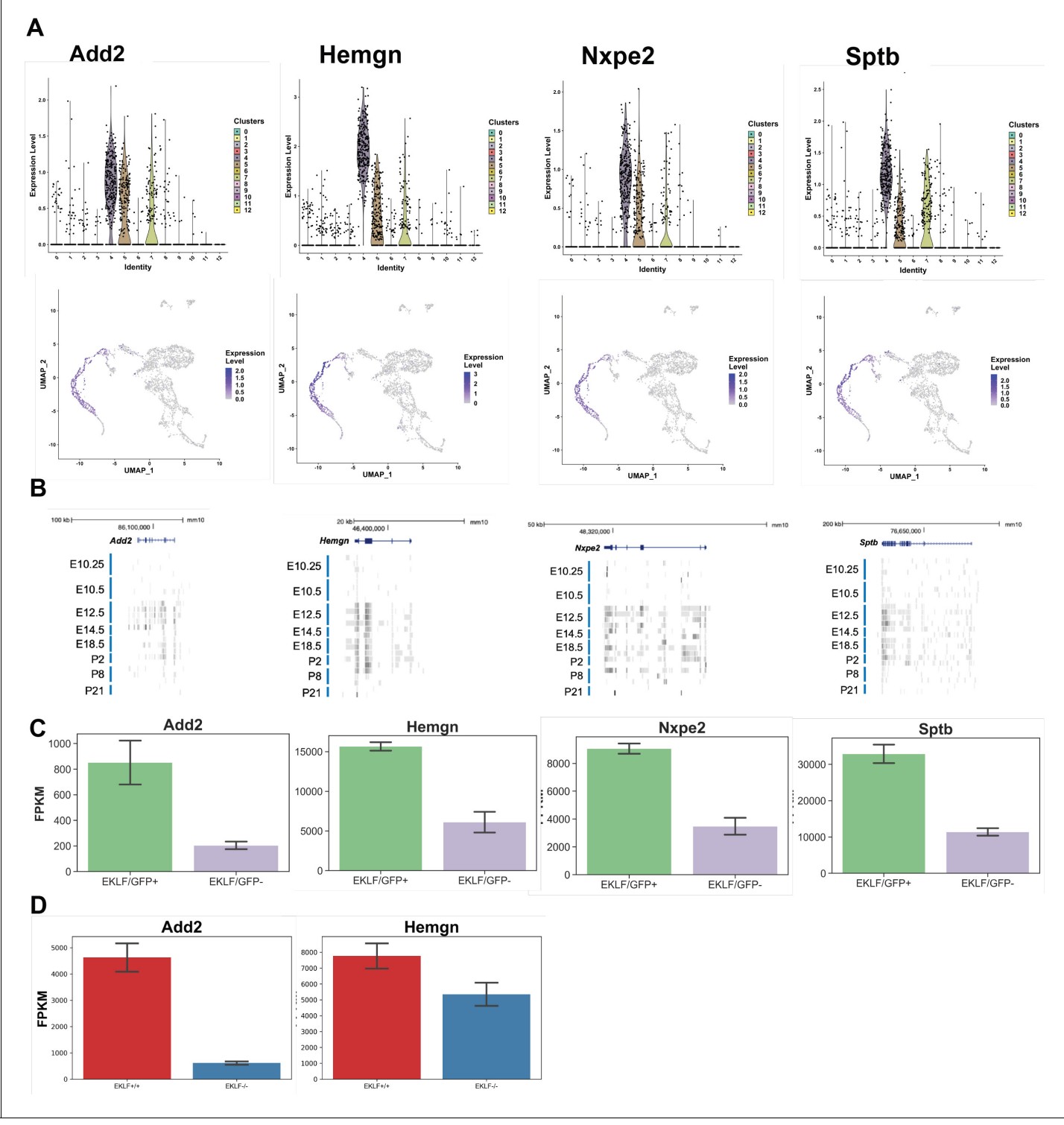

**Figure 8.** Identification of novel markers for F4/80+/EKLF+ fetal liver macrophages from single-cell sequencing. Using differential enrichment analysis of EKLF clusters 4, 5, and 7 compared with the rest of the cells, putative markers for F4/80+ EKLF+ cells were identified. (**A**) Violin and feature plots for the identified markers Add2 (adducinβ), Hemgn (hemogen), Nxpe2 (neurexophilin and PC-esterase domain family, member2), and Sptb (spectrinβ). (**B**) Data (as in *Figure 2C*, *Mass et al., 2016*) showing RNA-Seq reads of F4/80+ EKLF+ cell markers from staged and sorted fetal or postnatal liver macrophages. (**C**) FPKM expression levels of EKLF markers in F4/80+ EKLF/GFP+ and F4/80+ EKLF/GFP- fetal liver macrophage. (**D**) FPKM expression levels of EKLF markers Add2 and Hemgn in F4/80+ EKLF+/+ and F4/80+ EKLF-/- fetal liver macrophage.

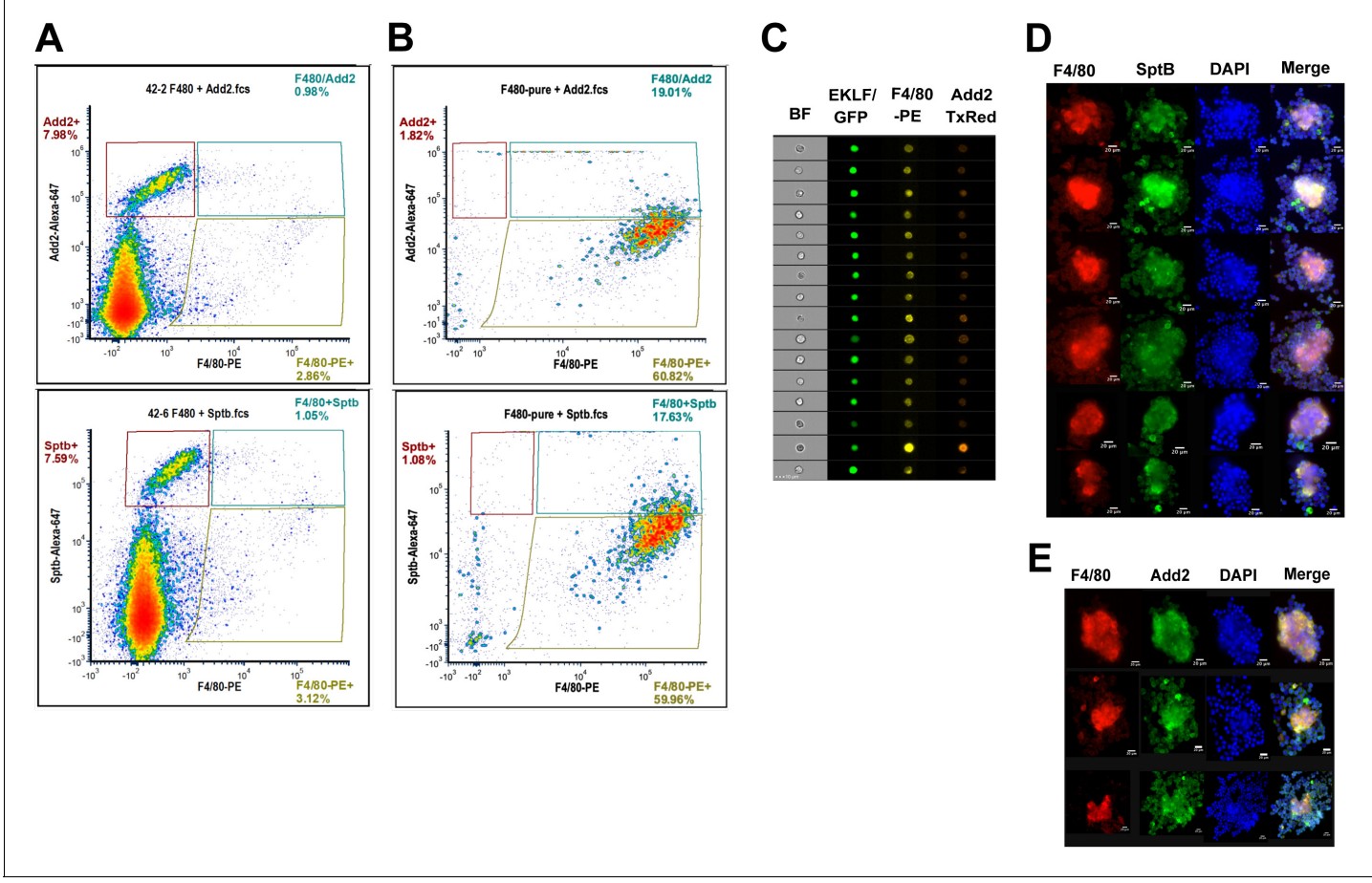

**Figure 9.** An improved strategy for antibody-based isolation of F4/80+/EKLF+ cells using novel markers identified from single-cell sequencing. (A) Flow cytometry analysis of E13.5 fetal liver cells stained with anti-F4/80-PE and anti-adducinβ (top) or anti-spectrinβ (below) antibodies conjugated to AlexaFluor 647. Gates are drawn based on unstained and single-color compensation controls for PE and AlexaFluor 647. Population percentages within each gate are indicated. (B) F4/80+ cells purified from E13.5 fetal livers using magnetic bead selection stained for anti-adducinβ (top) or anti-spectrinβ (below). Gates are the same as (A) and population percentages are indicated. (C) Imaging flow cytometry analysis of E13.5 fetal liver cells from the pEKLF/GFP mouse stained for F4/80-PE and Add2-TxRed. Single cells positive for F4/80, Add2, and GFP are shown. (D, E) Isolated erythroblast islands stained for DAPI, F4/80-PE, and (D) Sptb-Alexa647 or (E) Add2-Alexa647 and examined by fluorescent microscopy. Scale bars are indicated.

## EKLF regulation of island macrophage signature genes

By combining both EKLF-/- and EKLF/GFP+ RNA-Seq data, and then further parsed by the single-cell seq data, we find that loss of EKLF expression alters expression of many macrophage genes. We also find that the EKLF-expressing macrophages are functionally different from those not expressing EKLF, with high enrichment of genes performing functions consistent with erythroblast islands. Thus, the subset that are specific to the F4/80+ macrophage and whose expression is EKLF-dependent provides a novel expression signature that identifies targets that may be unique to the erythroblastic island. We have identified three in particular, Adra2b, Add2, and Sptb, that are enriched in EKLF WT macrophage and in the erythroblastic island. As a result, we suggest that these are additional novel markers that, in conjunction with F4/80, provide a further specification to island-associated macrophage identity. Heterogeneity remains an issue; however, from our single-cell seq data, it is likely that combining select markers, in particular F4/80+, Add2+, and Sptb+, will distinguish a discrete subpopulation that is highly enriched for island-associated macrophage. We are in the process of establishing such a protocol based on the robust cell surface expression of these three markers and will include Nxpe2 in the mix if a suitable selection antibody becomes available.

Identification and molecular knowledge of unique island macrophage expression and receptors may be functionally relevant to studies that utilize these cells to help expand in vitro erythropoiesis

more efficiently (*Hom et al., 2015*; *Rhodes et al., 2008*). These could be used in combination with cytokines known to enhance island macrophage such as erythropoietin (*Li et al., 2019*), dexamethasone (*Falchi et al., 2015*; *Heideveld et al., 2018*), or the KLF1-stimulated combo of ANGPTL7/IL33/SERPINB2 (*Lopez-Yrigoyen et al., 2019*). Efficient growth and maintenance become important when designing strategies to improve macrophage responses in the context of myelodysplastic syndromes (*Buesche et al., 2016*) or in the anemia of inflammation (*Hom et al., 2015*).

### Resolution of macrophage heterogeneity

Not surprisingly, we find that the FL F480+ population is heterogeneous, with our single-cell analysis suggesting 13 different clusters. Within this mixture we discovered a subset of clusters that express EKLF and its network of genes important for island macrophage. It is of interest that this subset does not express CD11b (Itgam), consistent with studies suggesting it is not an island macrophage marker (*Seu et al., 2017*; *Tay et al., 2020*; *Ulyanova et al., 2016*). Of additional interest, the granulocyte Ly6G marker did not appear in any of our clusters, consistent with an efficient removal of granulocytes during our enrichment procedure.

In this context, it is perhaps surprising that other markers historically suggested to be critical for island function such as Vcam1 are expressed at lower levels in the EKLF clusters than in others. Three explanations can be suggested. (1) EKLF+/Vcam1+ cells may be the relevant functional subset of total Vcam1-expressing cells, a different subset of which may have a separate, non-EKLF-dependent function (e.g., homing [(*Li et al., 2018*)]). (2) We are not suggesting that EKLF-expressing clusters are the sole source of macrophage islands; there may be others that arise following pathological conditions (e.g., ß-thalassemia or polycythemia vera [*Chow et al., 2013*; *Ramos et al., 2013*]), or when comparing steady-state versus stress/anemia (*Paulson et al., 2020*). (3) Erythroblastic islands are also found in bone marrow and spleen, and these arise within a significantly different niche than what we have focused on here during prenatal development. Such directive effects of the environment on macrophage identity have been noted before (*Gosselin et al., 2014*; *Lavin et al., 2014*). With respect to our present observations, given the importance of neural signaling in the bone marrow (*Méndez-Ferrer et al., 2020*), it is possible that a molecule such as Adra2b may be more highly expressed and play a more important role in bone marrow macrophage than in FL macrophage.

### Human island macrophage

Collectively, our study shows that EKLF plays a critical role within the specific subset of unique macrophage cells that are transiently required for proper establishment of erythroblastic islands in the developing embryo. Of relevance to human biology (*May and Forrester, 2020*), although the positive effects of EKLF expression on island macrophage function have been previously noted (*Lopez-Yrigoyen et al., 2019*), it is also relevant that a recent single-cell analysis of human FL hematopoiesis shows that EKLF and many of its target genes identified in the present study are also expressed in the 'Vcam1+ erythroblastic island macrophage' cluster (*Popescu et al., 2019*).

## Materials and methods

**Key resources table**

| Reagent type (species) or resource | Designation | Source or reference | Identifiers | Additional information |
|---|---|---|---|---|
| Genetic reagent (*Mus musculus*) | Klf$^{-/-}$ (Klf1$^{tm1Sho}$) | 10.1038/375318a0 | MGI:1857162 | EKLF-null mouse in 129S4/SvJae background |
| Genetic reagent (*Mus musculus*) | pEKLF/GFP | 10.1242/dev.018200 | Peklf-GFP | eGFP expressed from the EKLF promoter |
| Antibody | Anti-F4/80-PE (rabbit polyclonal) | eBiosciences | #12-4801-80 | (1:100) |
| Antibody | Anti-Adra2b (rabbit polyclonal) | Alomone Labs | #AAR-021 | (1:100) |
| Antibody | Anti-adducinβ (mouse monoclonal) | Santa Cruz Biotechnologies | # sc-376063 | (1:100) |

*Continued on next page*

*Continued*

| Reagent type (species) or resource | Designation | Source or reference | Identifiers | Additional information |
|---|---|---|---|---|
| Antibody | Anti-spectrinβ1 (mouse monoclonal) | Santa Cruz Biotechnologies | # sc-374309 | (1:100) |
| Antibody | Donkey anti-rabbit IgG – AlexaFluor 647 (donkey polyclonal) | Invitrogen | # A-31573 | (1:200) |
| Peptide, recombinant protein | FWV peptide (GenScript custom) | 10.1242/dev.103960 | # SC1848 | 2 mM |
| Commercial assay or kit | EasySep mouse PE positive selection kit | Cell Signaling Technologies | # 17656 | |
| Commercial assay or kit | Zip AlexaFluor 647 antibody labeling kit | Invitrogen | # Z11235 | |
| Commercial assay or kit | Lightning link Texas red conjugation kit | Abcam | # ab195225 | |
| Commercial assay or kit | RNA Nanoprep kit | Agilent | #400753 | |
| Commercial assay or kit | Chromium Single Cell 3' Library Kit v3 | 10X Genomics | # PN-1000095 | |
| Chemical compound | TRIzol reagent | Invitrogen | #15596026 | |
| Software, algorithm | STAR | 10.1093/bioinformatics/bts635 | RRID:SCR_015899 | |
| Software, algorithm | Salmon | 10.1038/nmeth.4197 | RRID:SCR_017036 | |
| Software, algorithm | HTSeq | 10.1093/bioinformatics/btu638 | RRID:SCR_005514 | |
| Software, algorithm | tximport | https://github.com/mikelove/tximport | RRID:SCR_016752 | |
| Software, algorithm | DESeq2 | 10.1186/s13059-014-0550-8 | RRID:SCR_015687 | |
| Software, algorithm | Alevin | 10.1186/s13059-019-1670-y | https://salmon.readthedocs.io | |
| Software, algorithm | Seurat | https://doi.org/10.1038/nbt.4096 | http://satijalab.org/seurat/ | |
| Software, algorithm | ggplot2 | https://github.com/tidyverse/ggplot2 | RRID:SCR_014601 | |
| Software, algorithm | Pandas | https://pandas.pydata.org | RRID:SCR_018214 | |
| Software, algorithm | Scikit-learn | http://scikit-learn.org/ | RRID:SCR_002577 | |
| Software, algorithm | Python Seaborn | https://seaborn.pydata.org/ | RRID:SCR_018132 | |
| Software, algorithm | Java Treeview | 10.1093/bioinformatics/bth349 | RRID:SCR_016916 | |
| Software, algorithm | Cluster 3.0 | 10.1093/bioinformatics/bth078 | RRID:SCR_013505 | |
| Software, algorithm | REViGO | http://revigo.irb.hr/ | RRID:SCR_005825 | |
| Software, algorithm | Generic GO Term Finder | 10.1093/bioinformatics/bth456 | RRID:SCR_008870 | |
| Software, algorithm | MEME-suite | http://meme-suite.org/ | RRID:SCR_001783 | |
| Software, algorithm | FCS Express 7 | https://www.denovosoftware.com | RRID:SCR_016431 | |

## Cell isolation

FLs were dissected from embryonic day E13.5 embryos and mechanically dispersed into single cells for fluorescence activated cell sorting (FACS) or RNA isolation. The heterozygous EKLF mouse strain was as described (*Perkins et al., 1995*). Photos were taken with a Nikon Microphot-FX fluorescence microscope equipped with a Q-Imaging camera or with a Zeiss Axio Observer Z1 equipped with a Hamamatsu C11440 camera. For single-cell sequencing, wild-type E13.5 FL cells were isolated from two littermate embryos from one donor mother, stained with anti-F4/80-PE antibody (eBiosciences #12-4801-80) and isolated using an EasySep mouse PE positive selection kit that uses a magnetic bead-based purification strategy (Cell Signaling Technologies #17656) and in the presence of 2 mM Icam4/αv inhibitor peptide (*Xue et al., 2014*) to eliminate macrophage–erythroid interactions. The cells were selected by repeating the magnetic bead binding step to increase purity. For immunofluorescence, erythroblastic island clusters were enriched from dispersed E13.5 FLs using a serum gradient as previously described (*Xue et al., 2014*).

## Flow cytometry

Suspended cells from FLs were stained for FACS with the following antibodies: anti-mouse F4/80-PE (eBiosciences #12-4801-80), anti-Adra2b (Alomone Labs #AAR-021), anti-adducinβ (Santa Cruz # sc-376063), and anti-spectrinβ1 (Santa Cruz # sc-374309). For anti-Adra2b staining, we used an Alexa 647 conjugated Donkey anti-rabbit secondary antibody (Life Technologies). For adducinβ and spectrinβ1 staining, primary unconjugated antibodies were conjugated to AlexaFluor 647 using a primary antibody conjugation kit (Invitrogen # Z11235). Flow cytometry data was analyzed by FCS Express software, and gates were drawn based on unstained and single-color compensation controls from the same samples, using the same dyes and within the same experiment.

## Imagestream analysis

Cells from intact E13.5 FLs were isolated from the pEKLF/GFP mouse and stained with the same antibodies for F4/80 and adducinβ as above, except the primary unconjugated antibody was labeled with a Texas Red labeling kit (Abcam #ab195225). Data was acquired using a Luminex Amnis Imagestream MkII Imaging Flow Cytometer and analyzed using the Amnis Ideas Software.

## Immunofluorescence

Erythroblastic island clusters were stained for F4/80 along with Add2 and Sptb using antibodies labeled as described above for flow cytometry. Photography was performed with a Nikon Microphot-FX fluorescence microscope equipped with a Q-Imaging camera.

## RNA isolation and RNA-Seq

FACS sorted cells were directly suspended in Trizol, and total RNA was extracted (*Rio et al., 2010*). RIN values for all EKLF+/+ and EKLF-/- samples were between 9.1 and 9.8. Poly-A library preparations of biological triplicate samples were analyzed by 100 nt single reads on an Illumina HiSeq 2500 or Illumina Novaseq, 60–90 million reads per sample. For F4/80+ EKLF/GFP+ population, the low cell numbers led us to use an Agilent RNA Nanoprep kit (#400753) for isolating reasonably good-quality RNA (RIN ~7). RNA-Seq data has been submitted to the Gene Expression Omnibus.

## Single-cell RNA-Seq

Libraries were generated from purified F4/80-PE+ using Chromium Single Cell 3' Reagent Kit V3 (10X Genomics) to generate cDNA and barcoded indexes for 25,000 individual cells. Paired-end sequencing was performed using a Novaseq instrument.

## Bioinformatics and computational analysis

*RNA-Seq* reads were aligned using STAR (*Dobin et al., 2013*) to the mouse genome (mm10) or mapped using Salmon (*Patro et al., 2017*) to the mouse transcriptome (Ensembl GRCm38). Htseq-count (*Anders et al., 2015*) was used to generate gene-specific raw counts from the STAR-aligned reads. Raw counts from these programs were imported using tximport package, and count normalization and differential gene expression analysis was performed using DESeq2 (*Love et al., 2014*). Hierarchical clustering and PCA (*Figure 1A, B*) were performed using R (http://www.R-project.org/) or Python Pandas (https://pandas.pydata.org) and Scikit-learn (https://scikit-learn.org, *Figure 3D*). All plots were generated using either R ggplot2 (https://ggplot2.tidyverse.org) or Python Seaborn (https://seaborn.pydata.org) and Plotnine (https://plotnine.readthedocs.io) libraries. k-means clustering was performed using Cluster 3.0 software (*de Hoon et al., 2004*), and heatmaps were generated using Java Treeview (*Saldanha, 2004*, *Figures 1D* and *3B*) and Python Seaborn (all others). R and Python code used in the analysis is deposited in github (https://github.com/mkaustav84/biekerlab-f480_macrophage; copy archived at swh:1:rev:907b15e74d998c5dd2a3106bce30af812c2b60b4.

*Single-cell sequencing* reads were aligned to the mouse transcriptome build GRCm38.p6vM24 using the software Alevin (*Srivastava et al., 2019*), and subsequent analysis was performed using the Seurat package (R-based) with built-in functions for plotting, clustering, PCA, and U-MAP analysis. After filtering, 3066 cells were retained, and for each cell 4000 variable genes were considered for analysis.

*Motif analysis* was performed using the Centrimo program (http://www.meme-suite.org/). Promoter sequences from −300 to +100 were extracted using a specific Perl script of Homer for the target EKLF-dependent gene set, and the promoters of the rest of the coding genes in the genome were used as background. GO analysis (go.princeton.edu) was performed using GO::TermFinder (*Boyle et al., 2004*), and GO terms were distilled using REVIGO (Jiang and Conrad similarity).

# Acknowledgements

We thank Chen-Yeh Ke and Ayan Ray (Soriano Lab) for advice on microscopy and the Sinai Genomics Technology Core for library preparation and deep sequencing of sorted RNA-Seq samples and single-cell sequencing. We thank Kristin Beaumont and Robert Sebra (Sinai Genomics Technology Core) for advice on single-cell sequencing analysis. Flow cytometry was performed at the Sinai Flow Cytometry Core. This work was supported by NIH grants R01 DK102260 and DK121671 to JJB, K01 DK115686 to MNG, and by a Black Family Stem Cell Institute Pilot Grant to KM.

# Additional information

## Funding

| Funder | Grant reference number | Author |
|---|---|---|
| National Institute of Diabetes and Digestive and Kidney Diseases | R01 DK102260 | James J Bieker |
| National Institute of Diabetes and Digestive and Kidney Diseases | R01 DK121671 | James J Bieker |
| National Institute of Diabetes and Digestive and Kidney Diseases | K01 DK115686 | Merlin Nithya Gnanapragasam |
| Black Family Stem Cell Institute | Postdoctoral award | Kaustav Mukherjee |

The funders had no role in study design, data collection and interpretation, or the decision to submit the work for publication.

## Author contributions

Kaustav Mukherjee, Data curation, Formal analysis, Investigation, Writing - original draft, Writing - review and editing; Li Xue, Investigation, Writing - review and editing; Antanas Planutis, Merlin Nithya Gnanapragasam, Data curation, Investigation, Writing - review and editing; Andrew Chess, Formal analysis; James J Bieker, Conceptualization, Supervision, Funding acquisition, Writing - original draft, Project administration, Writing - review and editing

## Author ORCIDs

Kaustav Mukherjee (iD) https://orcid.org/0000-0002-7057-0880
James J Bieker (iD) https://orcid.org/0000-0001-5128-7476

## Ethics

Animal experimentation: This study was performed in strict accordance with the recommendations in the Guide for the Care and Use of Laboratory Animals of the National Institutes of Health. All of the animals were handled according to approved institutional animal care and use committee (IACUC) protocols (#18-1911) of the Mount Sinai School of Medicine.

## Decision letter and Author response

Decision letter https://doi.org/10.7554/eLife.61070.sa1
Author response https://doi.org/10.7554/eLife.61070.sa2

## Additional files

### Supplementary files

• Supplementary file 1. Revigo analysis of the functions of fetal liver F4/80+ signature genes.

• Supplementary file 2. DESeq2 results of significantly downregulated genes in EKLF-/- cells.

• Supplementary file 3. DESeq2 results of significantly enriched genes in EKLF/GFP+ cells.

• Supplementary file 4. Complete results of Centrimo analysis of the promoters of EKLF-dependent genes in F4/80+ macrophages.

• Supplementary file 5. GO analysis of the top 100 differentially enriched genes in clusters 0, 1, 2, and 3 of the single-cell sequencing data.

• Supplementary file 6. GO analysis of the top 100 differentially enriched genes in clusters 4, 5, 7, and 8 of the single-cell sequencing data.

• Transparent reporting form

### Data availability

Data were deposited in GEO, accession number: GSE156153. Source data are included for Figures 1,3,4,5,6. R and Python code is deposited in https://github.com/mkaustav84/biekerlab-f480_macro-phage; copy archived at https://archive.softwareheritage.org/swh:1:rev:907b15e74d998c5dd2a3106bce30af812c2b60b4/.

The following dataset was generated:

| Author(s) | Year | Dataset title | Dataset URL | Database and Identifier |
|---|---|---|---|---|
| Mukherjee K, Planutis A, Xue L, Bieker JJ | 2021 | EKLF/Klf1 expression specifies a unique macrophage subset during mouse erythropoiesis | https://www.ncbi.nlm.nih.gov/geo/query/acc.cgi?acc=GSE156153 | NCBI Gene Expression Omnibus, GSE156153 |

The following previously published dataset was used:

| Author(s) | Year | Dataset title | Dataset URL | Database and Identifier |
|---|---|---|---|---|
| Lavin Y, Winter D, Blecher-Gonen R, David E, Keren-Shaul H, Merad M, Jung S, Amit I | 2014 | Tissue-resident macrophage enhancer landscapes are shaped by the local microenvironment | https://www.ncbi.nlm.nih.gov/geo/query/acc.cgi?acc=GSE63340 | NCBI Gene Expression Omnibus, GSE63340 |

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
