## [Decision Letter]

**Acceptance summary:**

Your study characterised thru single cell RNA-sequencing the various murine macrophages population present in the fetal liver. Importantly, this study identified and characterised thoroughly a new population of EKLF expressing / KLF1 dependent macrophages associated with erythroid islands. This work is important in the field and should aid future studies on macrophage development and identity, as well as erythropoiesis.

**Decision letter after peer review:**

Thank you for submitting your article "EKLF/KLF1 expression defines a unique macrophage subset during mouse erythropoiesis" for consideration by *eLife*. Your article has been reviewed by two peer reviewers, and the evaluation has been overseen by a Reviewing Editor and Carla Rothlin as the Senior Editor. The following individual involved in review of your submission has agreed to reveal their identity: Christian Schulz (Reviewer #2).

The reviewers have discussed the reviews with one another and the Reviewing Editor has drafted this decision to help you prepare a revised submission.

Summary:

The authors use scRNA-seq to identify distinct macrophage populations in mouse E13.5 fetal liver. They focus on the description of a small subset of macrophages associated with erythroid islands that expresses EKLF. They further determined EKLF-dependent gene expression programs using mice deficient of this transcription factor and provided new surface markers that can be used to isolate these cells via flow cytometry. This work is important in the field and should aid future studies on macrophage development, identity and erythropoiesis.

Essential revisions:

Although interesting, some conclusions drawn from the data should be further supported experimentally and further validation should be provided on 4 points:

1) The RNAseq performed on EKLF-/- and WT FL macrophages is analyzed and interpreted in a way that macrophages, despite reduced numbers, are still just one population of cells, although the authors indicate beforehand that only 36% express EKLF. The differential expression of genes observed here could be simply due to the fact that EKLF+ macrophages don't survive, and instead Vcam1+ and Epor+ macrophages (or other macrophage subpopulations) relatively increase in numbers. A possible further heterogeneity should be considered before the interpretation of "EKLF-dependent genes" is valid (see also point 4). Thus, statements such as “Critically, EKLF regulates the expression of a significant number of other transcription factors in FL

macrophages including Foxo3, Ikzf1, MafK, Nr3c1; cell-cycle E2f factors; and other members of the Klf family (Figure 5—figure supplement 1C). Thus, along with the known critical transcriptional role of EKLF in erythroid cells, our data suggest a crucial global regulatory role for EKLF in proliferation and development of FL island macrophages.” are highly speculative, particularly if genes do not possess an EKLF binding site in their enhancer. Similarly, genes in EKLF-/- cells may be downregulated simply due to the absence of these cells (data shown in Figure 6B).

2) Validation of RNA-seq (and observed heterogeneity) on protein level in the intact FL tissue (not isolated cells only):

– Heterogeneity of scRNA-seq analysis has to be confirmed in a protein level, with a panel consisting of at least F4/80, EKLF(-GFP), Adra2b, Add2, Sptb2, Vcam1. Analysis should be performed in an unbiased manner (similar to scRNA-seq using UMAP)

– EKLF expression is shown for fetal liver cells (Figure 2A, B). The authors should perform IF analyses on cryosections so that the reader can appreciate the co-localization with central macrophages and EKLF expression in situ.

– Along these lines, an IF should be performed on cryosections to show that EKLF+ macrophages are indeed also Vcam1 and/or Epor. The scRNA-seq data indicates that "most EKLF+ cells express Epor", but Epor is expressed by various other clusters. Quantification of these markers in situ would allow to address whether these cells are really the same or whether a further heterogeneity exists.

– to make sure that erythroid markers (Add2, Sptb) are indeed not contaminants, imagestream should be performed. Similar to CD31 "expression" in the adult liver where they are just contaminated with endothelial cell parts (Lynch et al., 2018), macrophages may be just attached to membrane parts of erythroblasts that are being stained.

– Only a very minor fraction of cluster 4 expresses Adra2b. What does that mean for conclusion in Figure 6D? Are all the other macrophages (e.g. cluster 0/1) not erythroblastic islands during normal hematopoiesis in the FL? “Second, we are not suggesting that EKLF-expressing clusters are the sole source of macrophage islands…” suggest exactly this; however, data are lacking to confirm this statement (e.g. IF staining of Vcam1/Marco + Ter119).

3) Function of EKLF macrophage by better describing the knock-out phenotype:

What is the phenotype of EKLF-/- embryos?

Showing data as % (Figure 6 C) does not allow a conclusion of how the populations change in the KO. These numbers have to be quantified as cells per liver or per gram of tissue. Why are not other marker used for this analysis that would allow enrichment of FL macrophages?

Figure 3 A does not show differences in F4/80 macrophages between wildtype and EKLF-/- FL. Further, the gating for macrophages should not be done solely on F4/80 expression, at least CD45 and CD11b should be added to the gating strategy.

Are EKLF-expressing F4/80+ FL macrophages functionally different to EKLF-negative macs?

4) As mentioned and demonstrated by the authors, FL macrophages are heterogenous. One factor contributing to this heterogeneity is the developmental origin of macrophages, since both AGM and yolk sac provide hematopoietic cells accumulate in the fetal liver. While this paper focused on FL macrophages, authors should check whether key markers are expressed in YS erythroblastic islands. This would increase our understanding regarding FL-specificity of their findings.

[Editors' note: further revisions were suggested prior to acceptance, as described below.]

Thank you for submitting your article "EKLF/KLF1 expression defines a unique macrophage subset during mouse erythropoiesis" for consideration by *eLife*. Your article has been reviewed by two peer reviewers, and the evaluation has been overseen by a Reviewing Editor and Carla Rothlin as the Senior Editor. The following individual involved in review of your submission has agreed to reveal their identity: Christian Schulz (Reviewer #2).

The reviewers have discussed the reviews with one another and the Reviewing Editor has drafted this decision to help you prepare a revised submission.

Summary:

The authors used scRNA-seq to identify distinct macrophage populations in the E13.5 FL. Using previously published data in combination with newly performed RNA-seq analysis, they focus on the description of a small subset of macrophages that expresses EKLF. The authors present data of EKLF KO mice, indicating EKLF target genes and new surface markers that can be used to isolate these cells via flow cytometry.

Essential revisions:

To consider the manuscript for publication, the authors need to include a clear validation of heterogeneity on the protein level either by flow cytometry or 4-color immunofluorescence of the intact fetal liver.

[Editors' note: further revisions were suggested prior to acceptance, as described below.]

Thank you for resubmitting your work entitled "EKLF/KLF1 expression defines a unique macrophage subset during mouse erythropoiesis" for further consideration by *eLife*. Your revised article has been evaluated by Carla Rothlin (Senior Editor), a Reviewing Editor and the reviewers.

The manuscript has been improved but there are some remaining issues that need to be addressed before acceptance, as outlined below:

We all concur on the important contributions of your manuscript to the field. We are also in agreement that this work will inspire and enable future studies on macrophage heterogeneity and function during development. Establishing a protocol to isolate these cells based on cell surface markers will strengthen the findings. The stainings included in response to our request have not been extensively confirmed and therefore weaken what is otherwise an important contribution.

Since these data are not sufficiently validated, we request not to include them in the final paper. However, the community should learn from your efforts in establishing this protocol and the fact that we all consider this to be of great importance.

As such, we request that:

– While it would be ideal to have had established a protocol for cell isolation, the stainings included in the revision have not been extensively validated and should not be included in the manuscript.

– The authors should also discuss their efforts in establishing a protocol to isolate these cells based on the cell surface expression of Add2, Nxpe2 and Sptb without adequate success and that future efforts are warranted to establish a solid method for the isolation of this population.

– We expect that the authors will continue their efforts in establishing this approach and that they report them as a preprint on bioRxiv or if appropriate as a Research Advance in *eLife*, either of which would be linked to the original paper.

---

## [Author Response]

Essential revisions:Although interesting, some conclusions drawn from the data should be further supported experimentally and further validation should be provided on 4 points:1) The RNAseq performed on EKLF-/- and WT FL macrophages is analyzed and interpreted in a way that macrophages, despite reduced numbers, are still just one population of cells, although the authors indicate beforehand that only 36% express EKLF.

We are not considering macrophage as one population of cells; rather, we are focusing on the F480+ subpopulation based on prior knowledge of the subset that plays a role in erythroblastic islands. These are further subdivided to focus on the 1/3 that expresses EKLF. A major point of the paper is acknowledgement of macrophage heterogeneity, culminating in our single cell seq analysis that indicates an extensive range of genotypes even within the F480+ population.

The differential expression of genes observed here could be simply due to the fact that EKLF+ macrophages don't survive, and instead Vcam1+ and Epor+ macrophages (or other macrophage subpopulations) relatively increase in numbers.

The fact that there are EKLF+ and F480+ double positive macrophage by two criteria (Figures 2 and 4) makes it clear that survival is not affected by presence/absence of EKLF. Hence, we do not believe that EKLF+ macrophage don’t survive.

A possible further heterogeneity should be considered before the interpretation of "EKLF-dependent genes" is valid (see also point 4). Thus, statements such as “Critically, EKLF regulates the expression of a significant number of other transcription factors in FLmacrophages including Foxo3, Ikzf1, MafK, Nr3c1; cell-cycle E2f factors; and other members of the Klf family (Figure 5—figure supplement 1C). Thus, along with the known critical transcriptional role of EKLF in erythroid cells, our data suggest a crucial global regulatory role for EKLF in proliferation and development of FL island macrophages.” are highly speculative, particularly if genes do not possess an EKLF binding site in their enhancer.

We have modified the statement in question that suggest EKLF targets by the dual-RNA seq criteria by stating that these can be verified by a search of consensus target and ultimately by ChIP. The best way to directly address EKLF-dependent genes would be first comparing single cell RNA-Seq data of WT and EKLF-/- macrophages followed by EKLF ChIP-seq or Cut&Run in F4/80+ EKLF+ macrophages isolated using the Add2/Sptb method.

However, it appears we have not been clear about our methodology. Our statements/conclusions are based on two different experiments, one comparing RNA seq from EKLF WT vs null, and the other comparing RNA seq in WT cells segregated into those that do or do not express EKLF/GFP. Overlapping these two independent datasets is an extremely powerful way to parse down the potential direct/indirect genes whose expression are dependent on the presence of EKLF. We have clarified this point in the revision.

Similarly, genes in EKLF-/- cells may be downregulated simply due to the absence of these cells (data shown in Figure 6B).

As a corollary to our earlier statement, the fact that by two criteria (Figures 2 and 4) there are F480+ macrophage that are EKLF- makes it clear that survival is not affected by presence/absence of EKLF. Hence, we do not believe that EKLF-/- cells are absent. Examination of the data suggests the following. First, in Figure 5A we would expect the vast majority of genes enriched in EKLF/GFP+ (co-expressed in EKLF+ macrophages) to be downregulated in EKLF-/- i.e. the vast majority of data points would lie in the top left quadrant, with a high overall correlation between EKLF/GFP+ enrichment and EKLF-/- downregulation. However, we see a fairly high number of genes in the top right quadrant and these are genes enriched in EKLF/GFP+ but also upregulated in EKLF-/-. Further, the overall correlation in the two datasets is only a modest -0.11. Second, our data shows that about 25-35% of F4/80+ cells on average maybe EKLF+ (Xue et al., 2014, Figure 7A). Thus, we would expect the reduction of F4/80+ cells in EKLF-/- to be around the same percentage if they do not survive in EKLF-/-. However, although our data shows a 50% reduction in F4/80+ macrophages in EKLF-/- (Figure 3A) compared to WT (Figure 1—figure supplement 1A).

For the future, to investigate the cause of the reduced survival of F4/80+ cells in the EKLF-/- fetal liver, we are establishing a conditional knockout model where EKLF is knocked out only in EKLF-expressing F4/80+ macrophages in order to delineate the erythroid and macrophage roles of EKLF.

2) Validation of RNA-seq (and observed heterogeneity) on protein level in the intact FL tissue (not isolated cells only):– Heterogeneity of scRNA-seq analysis has to be confirmed in a protein level, with a panel consisting of at least F4/80, EKLF(-GFP), Adra2b, Add2, Sptb2, Vcam1. Analysis should be performed in an unbiased manner (similar to scRNA-seq using UMAP)

This is an interesting suggestion, so we attempted these studies using a mix of antibodies. We used total (ie, unsorted) E13.5 FL cells from EKLF/GFP mouse and stained for F4/80, Add2, Sptb, Vcam1, Adra2b using primary conjugated antibodies with nonoverlapping fluors. Unfortunately, we ran into problems/ limitations. Even in single antibody stained compensation controls, very few Adra2b stained cells were observed after conjugating with an Alexa Fluor 700 labeling kit from Abcam (ab269824) and the anti Vcam1AlexaFluor405 antibody (SCBT) did not work (ie, no Vcam1+ cells were detected – highly unlikely from intact fetal liver). Further, the conjugation of the primary Add2 antibody with a Texas-red labeling kit from Abcam led to greatly diminished signals than was observed with a different labeling kit in Figure 9A,B. Nonetheless, analysis of of 4 markers using tSNE show significant segregation and thus heterogeneity consistent with the RNA seq data. Even though we have two independent data sets that support this idea, we hesitate to include these data in the manuscript, as we consider it incomplete. Thus, it is still a work in progress that will require optimization and rigorous testing of various antibodies and labeling kits. Though we agree that this data would enhance our conclusions, the lack of this data does not detract from our original findings. Therefore, we request that the editor and reviewers take this into consideration and excuse our inability to perform this experiment at this time. We recognize the importance of correlating the observed heterogeneity in single cell RNA-Seq to protein expression, and we are currently endeavoring to investigate this.

– EKLF expression is shown for fetal liver cells (Figure 2A, B). The authors should perform IF analyses on cryosections so that the reader can appreciate the co-localization with central macrophages and EKLF expression in situ.– Along these lines, an IF should be performed on cryosections to show that EKLF+ macrophages are indeed also Vcam1 and/or Epor. The scRNA-seq data indicates that "most EKLF+ cells express Epor", but Epor is expressed by various other clusters. Quantification of these markers in situ would allow to address whether these cells are really the same or whether a further heterogeneity exists.

We had performed cryosection analyses before submission of the original manuscript, but felt them uninformative for two reasons. One, the (beautiful) structure of the island macrophage (ie, F4/80+) spilled into adjacent sections, making it difficult to ascertain differences between WT vs EKLF-null fetal livers. Two, unlike our FACS studies, the EKLF/GFP signal was weak in the cryosections and did not provide a suitable way to distinguish/quantify its presence in the clusters of cells. We feel the single cell IF analysis of Figure 2 is the most appropriate way to directly address EKLF protein presence in the F480+ macrophage. Complementing this, our new data of Figure 9D,E show the colocalization of F480+ cells with SptB or Add2 in isolated islands, and image stream analyses (new Figure 9C) show overlap of EKLF/GFP, F480+ and Add2 in single cells. Specifically, we isolated erythroblastic islands using E13.5 fetal livers using a serum gradient and stained for F4/80 along with Add2 and Sptb. We have added the relevant descriptions of our results in the revised manuscript.

– To make sure that erythroid markers (Add2, Sptb) are indeed not contaminants, imagestream should be performed. Similar to CD31 "expression" in the adult liver where they are just contaminated with endothelial cell parts (Lynch et al., 2018), macrophages may be just attached to membrane parts of erythroblasts that are being stained.

We performed Imagestream analysis on E13.5 fetal liver cells isolated from the EKLF/GFP mouse and stained with primary conjugated antibodies for F4/80 and Add2. The results are shown in Figure 9C and relevant text descriptions are added in the revised manuscript. We are not sure if the reviewer means that the Sptb/Add2 staining is coming from erythroid membranes attached to macrophages (interaction disrupted but membrane fragments remain attached) or erythroid cells attached to macrophages. Our Imagestream result clearly rules out the latter, but the former is also unlikely since we also observe the Add2/Sptb mRNA in single cell RNA-Seq of macrophages.

– Only a very minor fraction of cluster 4 expresses Adra2b. What does that mean for conclusion in Figure 6D? Are all the other macrophages (e.g. cluster 0/1) not erythroblastic islands during normal hematopoiesis in the FL? “Second, we are not suggesting that EKLF-expressing clusters are the sole source of macrophage islands…” suggest exactly this; however, data are lacking to confirm this statement (e.g. IF staining of Vcam1/Marco + Ter119).

Our interpretation is that the few Adra2b+ macrophages in Cluster4 are island macrophages, but that does not suggest that there are no other clusters of island macrophages. We have tried to focus on the role of EKLF in fetal liver macrophages, and to that end found Adra2b as a possible marker for a small subset of them. Directly observing that this marker is also expressed in island macrophages suggest that at least this subset of EKLF expressing macrophages are associated with erythroblastic islands. Our data with Add2 and Sptb is more robust than Adra2b, as these stain 26% of F480+ cells and are present in island macrophages.

3) Function of EKLF macrophage by better describing the knock-out phenotype:What is the phenotype of EKLF-/- embryos?

We inadvertently left out this critical information. EKLF-/- embryos are embryonic lethal at E15. At E13.5 they are severely anemic and their pale fetal liver is already distinct compared to their EKLF+/+ and EKLF+/- littermates. Given our studies, the cause of the anemia could be a combination of impaired erythropoiesis due to the loss of EKLF in developing erythroid progenitors as well as impaired island macrophage function supporting definitive erythropoiesis. This information has been added to the Introduction and to the Discussion.

Showing data as % (Figure 6 C) does not allow a conclusion of how the populations change in the KO. These numbers have to be quantified as cells per liver or per gram of tissue. Why are not other marker used for this analysis that would allow enrichment of FL macrophages?

In Figure 6C, we have used F4/80 as a marker which specifically shows the fetal liver macrophage population. The percentages indicated already accounting for the total cells per liver since intact E13.5 fetal livers were used for this analysis. Other markers are not used since it has been shown previously that F4/80 is specifically expressed on fetal liver macrophages, unlike Mac1 or Cd11b for example; hence, we focused on a positively-associated marker for our studies.

Figure 3 A does not show differences in F4/80 macrophages between wildtype and EKLF-/- FL. Further, the gating for macrophages should not be done solely on F4/80 expression, at least CD45 and CD11b should be added to the gating strategy.

The population of F4/80+ cells in WT and KO are shown in Figure 1—figure supplement 1A and Figure 3A respectively. About 10% of E13.5 fetal liver cells in WT are F4/80+ (Figure 1—figure supplement 1A) and that is reduced to 5% in EKLF-/- (Figure 3A). CD11b is not expressed in fetal liver island macrophages so it was not included. Further, the data from (Mass et al., 2016) already includes CD45 in their gating strategy and we find that EKLF is expressed in the FL F4/80+ CD45+ macrophages in their dataset (Figure 2C).

Are EKLF-expressing F4/80+ FL macrophages functionally different to EKLF-negative macs?

Yes. When we perform GO analysis on the genes enriched in F4/80+ EKLF/GFP+ macrophages we find that these genes are mostly involved in erythroid and myeloid development, heme biosynthesis, and iron transport. In contrast the genes enriched in F4/80+ EKLF/GFP- macrophages are involved in immune system processes such as leukocyte and lymphocyte activation, and innate immunity.

We apologize for excluding this analysis from our initial submission – we recognize the importance of this data in emphasizing the unique nature of EKLF+ macrophages, which is the core idea our study is trying to convey, and hence this has been added to Table 2 and Table 3, and described in the revised manuscript. Additionally, we have also included a summary of significant GO terms associated with genes downregulated in EKLF-/- macrophages in Table 1 and described in the revised manuscript.

4) As mentioned and demonstrated by the authors, FL macrophages are heterogenous. One factor contributing to this heterogeneity is the developmental origin of macrophages, since both AGM and yolk sac provide hematopoietic cells accumulate in the fetal liver. While this paper focused on FL macrophages, authors should check whether key markers are expressed in YS erythroblastic islands. This would increase our understanding regarding FL-specificity of their findings.

The yolk sac does not have erythroblastic islands. Perhaps there is a confusion of terminology, as yolk sacs have ‘blood islands’, which are not the same, as these are morphologically-defined bulges in the vascular system that are filled with erythroid cells. Phenotypically, the yolk sac is home to the first two waves of hematopoiesis, with the first yielding primitive erythroid cells (starting at E7.25), and the second yielding erythro-myeloid progenitors (EMPs) that arise at E8.25, which can be defined as CD45-/ckit+/F480-. Our previous study showed that EKLF is expressed in these EMPs, and in the CD45+/ckit-/F480+progeny, observations also seen in the Mass et al. study. Single cell analysis of these cells would be of high interest, but is not within the scope of this study. A second study to address the subsequent fate of the EKLF+ EMP cell would be lineage tracing; unfortunately, that requires an EKLF-CRE/ER mouse, a line not yet available.

However, the AGM would be quite interesting to analyze in the context of developmental origins, as pointed out by the reviewer. Currently, we are attempting to establish lineage tracing mouse models to answer such questions.

[Editors' note: further revisions were suggested prior to acceptance, as described below.]

Essential revisions:To consider the manuscript for publication, the authors need to include a clear validation of heterogeneity on the protein level either by flow cytometry or 4-color immunofluorescence of the intact fetal liver.

We used total (ie, unsorted) E13.5 FL cells from EKLF/GFP mouse and stained for F4/80, Add2, and Sptb, using primary conjugated antibodies with nonoverlapping fluors, and have included two independent data sets. Analysis of these 4 fluorescent markers using tSNE shows significant segregation and thus validate heterogeneity consistent with the RNA seq data, and super-positioning of these single-cell graphs demonstrate both distinct and overlapping expression clusters (described in the text and included in Figure 9).

[Editors' note: further revisions were suggested prior to acceptance, as described below.]

The manuscript has been improved but there are some remaining issues that need to be addressed before acceptance, as outlined below:We all concur on the important contributions of your manuscript to the field. We are also in agreement that this work will inspire and enable future studies on macrophage heterogeneity and function during development. Establishing a protocol to isolate these cells based on cell surface markers will strengthen the findings. The stainings included in response to our request have not been extensively confirmed and therefore weaken what is otherwise an important contribution.Since these data are not sufficiently validated, we request not to include them in the final paper. However, the community should learn from your efforts in establishing this protocol and the fact that we all consider this to be of great importance.As such, we request that:– While it would be ideal to have had established a protocol for cell isolation, the stainings included in the revision have not been extensively validated and should not be included in the manuscript.– The authors should also discuss their efforts in establishing a protocol to isolate these cells based on the cell surface expression of Add2, Nxpe2 and Sptb without adequate success and that future efforts are warranted to establish a solid method for the isolation of this population.– We expect that the authors will continue their efforts in establishing this approach and that they report them as a preprint on bioRxiv or if appropriate as a Research Advance in eLife, either of which would be linked to the original paper.

Thank you for the positive comments on the importance of the manuscript. As requested, we have removed the recently added data, and have indicated that we are establishing/optimizing robust protocols using F4/80, Add2, Sptb, and Nxpe2 to isolate erythroblastic island populations. We will update the research community as soon as we are successful.